# Automatically Identify and Rectify: Robust Deep Contrastive Multi-view Clustering in Noisy Scenarios

**Xihong Yang** [1 2]  **Siwei Wang** [3]  **Fangdi Wang** [1]  **Jiaqi Jin** [1]
**Suyuan Liu** [1]  **Yue Liu** [2]  **En Zhu** [1]  **Xinwang Liu** [1]  **Yueming Jin** [2]

## Abstract

Leveraging the powerful representation learning capabilities, deep multi-view clustering methods have demonstrated reliable performance by effectively integrating multi-source information from diverse views in recent years. Most existing methods rely on the assumption of clean views. However, noise is pervasive in real-world scenarios, leading to a significant degradation in performance. To tackle this problem, we propose a novel multi-view clustering framework for the automatic identification and rectification of noisy data, termed AIRMVC. Specifically, we reformulate noisy identification as an anomaly identification problem using GMM. We then design a hybrid rectification strategy to mitigate the adverse effects of noisy data based on the identification results. Furthermore, we introduce a noise-robust contrastive mechanism to generate reliable representations. Additionally, we provide a theoretical proof demonstrating that these representations can discard noisy information, thereby improving the performance of downstream tasks. Extensive experiments on six benchmark datasets demonstrate that AIRMVC outperforms state-of-the-art algorithms in terms of robustness in noisy scenarios. The code of AIRMVC are available at https://github.com/xihongyang1999/AIRMVC on Github.

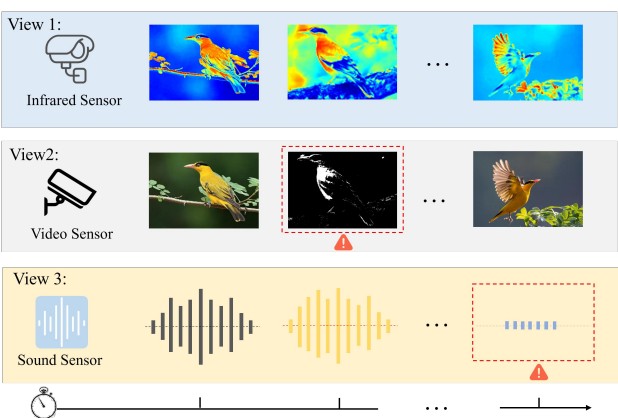

*Figure 1.* An illustrative diagram of noise in a multi-view scenario. In the diagram, the areas marked with red exclamation points indicate instances where sensor failures or malfunctions at specific moments lead to data corruption. Compared to other views, these instances are considered noisy data.

## 1. Introduction

In real-world scenarios, multi-source information data is prevalent. To effectively handle such data, Multi-View Clustering (MVC) has emerged as a powerful unsupervised method. In recent years, MVC has gained significant attention and has become a prominent focus of research. Existing MVC methods can be broadly categorized into two main groups. The first group comprises traditional approaches, including Multiple Kernel Clustering (MKC) (Liu et al., 2021), Non-negative Matrix Factorization (NMF) (Wen et al., 2018), subspace clustering(Zhou et al., 2019), and graph-based clustering (Liu et al., 2022a). With the advancement of deep neural networks, deep multi-view clustering algorithms (Yang et al., 2023b; Sun et al., 2024), representing the second category of MVC methods, have garnered significant attention from researchers.

While existing MVC algorithms have demonstrated notable clustering performance, they predominantly rely on the assumption that the features from all views are clean. These approaches generally adopt a standard workflow: representations are first extracted via an encoder, followed by a

---

[1]School of Computer, National University of Defense Technology, Changsha, Hunan, China [2]National University of Singapore, Singapore. [3]Intelligent Game and Decision Lab, Beijing, China, xihong_edu@163.com. Correspondence to: Xinwang Liu <xinwangliu@nudt.edu.cn>.

*Proceedings of the 42nd International Conference on Machine Learning*, Vancouver, Canada. PMLR 267, 2025. Copyright 2025 by the author(s).

feature fusion strategy, and then utilized for downstream clustering tasks. The integration of clean views enables the exploitation of complementary information from different perspectives of the same sample, resulting in enhanced performance compared to single-view clustering methods. This complementary information across views is instrumental in revealing the underlying cluster structures within the data. How can the complementary information across multiple views be effectively captured? Contrastive learning presents a paradigm that utilizes self-supervised techniques to learn cross-view consistency, ensuring coherent predictions across diverse views (Lu et al., 2024; Yang et al., 2023b; Xu et al., 2022b). Alternatively, self-training explores consistency by generating a unified cluster partition, thereby facilitating the discovery of complementary information (Xu et al., 2022a; Wang et al., 2021).

Although the aforementioned methods have demonstrated promising results, we identified notable limitations when applied to real-world scenarios involving noisy data. Fig. 1 illustrates an example of noisy data in a multi-view setting, where three views are presented, and random noise exists within some of them. Moreover, we observed a significant performance degradation in existing methods under such noisy conditions (Tab. 4). The noisy data not only fail to contribute positively during multi-view feature fusion but also disrupt the underlying cluster structures. Moreover, the erroneous influence of noisy data introduces considerable bias in the optimization process, thereby diminishing the advantages of complementary information across views. Consequently, the performance of these methods may even fall below that of single-view models. Detailed experimental evidence supporting this claim is presented in Section. 5.3. Recently, some noisy-based deep multi-view clustering methods have been proposed to alleviate the problem. RMCNC (Sun et al., 2024) designed a noise-tolerance contrastive loss to mitigate the impact for noisy correspondence. MVCAN (Xu et al., 2024) employed un-shared network structure and designed a two-level optimization for multi-view clustering. Although a large improvement has been made, the exploration of noisy data in those methods remains focused on enhancing feature robustness. However, they have yet to develop dedicated frameworks for the identification and rectification of noisy data.

To automatically identify and rectify the noisy data in multi-view scenario, we propose a novel deep contrastive multi-view clustering framework, termed **AIRMVC**. Specifically, we reformulate noise identification as an anomaly identification problem and introduce Gaussian Mixture Model (GMM) to address this problem. By substituting the latent variable of GMM with the soft prediction, we enable a dynamic update of GMM parameters. Based on the identification results, we introduce a hybrid rectification strategy that employs an interpolation mechanism to alleviate the ad-

verse effects of noisy data, thereby enhancing the robustness of the correction process. In this way, the adverse effects of noisy data could be mitigated. Furthermore, by carefully refining the soft predicted distributions, we design a noise-robust contrastive mechanism to enhance the discriminative capacity of the learned representations. Furthermore, we conduct a theoretical investigation of the contrastive mechanism to validate the stability and robustness of the learned representations. Extensive experiments conducted on six benchmark datasets demonstrate the effectiveness and robustness of our proposed method. The key contributions of our paper are summarized as follows:

- We reformulate the noise identification as an anomaly identification problem, solving it by GMM. Based on the results, we propose a hybrid rectification strategy to automatically correct the noisy data.

- A noise-robust contrastive mechanism is proposed to generate more reliable representations. Besides, we theoretically prove that the generated representations could discard noisy information to benefit the downstream task.

- We conduct extensive experiments on six benchmark datasets to verify the effectiveness and robustness of AIRMVC.

## 2. Preliminary & Problem Definition

In this paper, we focus on the multi-view clustering task with noisy inputs, as noise is a common occurrence in multi-source information inputs in real-world scenarios. To address this challenge, we aim to enhance the robustness of the network by automatically identifying and rectifying noisy data in an unsupervised multi-view setting. For simplicity, we provide the following symbolic definitions. Given a dataset $\{x^v\}_{v=1}^V$ with $V$ views and $N$ samples, we define $\mathbf{E}_i^v$ and $y_i^v$ as the extracted representations and the soft predictions of class probabilities for each sample, respectively. The encoder network and decoder network are denoted as $\mathcal{F}^v(x^v; \Theta^v)$ and $\mathcal{G}^v(\mathbf{E}^v; \Phi^v)$, respectively. Additional notations are summarized in Tab. 5 in the Appendix.

Most existing multi-view clustering (MVC) methods operate under the assumption that the input multi-view data is both complete and consistent. In such idealized scenarios, the primary learning paradigm in MVC involves learning representations for $V$ independent views, followed by the application of various fusion strategies to effectively extract and utilize shared information across the views. Although these models demonstrate strong performance under ideal conditions, we observe a significant decline in robustness when low-quality, noisy views are introduced, leading to substantial performance degradation. However, noise is a

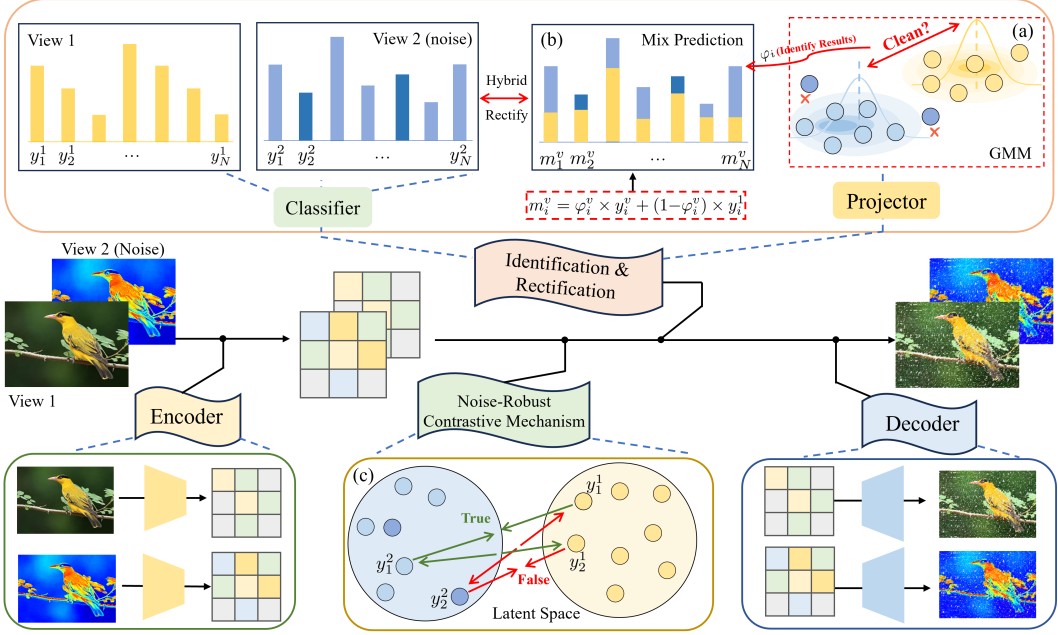

*Figure 2.* Illustration of the overall framework of the proposed AIRMVC. Specifically, we first encode the input multi-view data to generate representations. Next, an automatic noise identification and rectification strategy is introduced to mitigate the adverse impact of noisy data. Simultaneously, we propose a noise-robust contrastive mechanism to generate more reliable and discriminative representations for the downstream clustering task.

pervasive issue in real-world scenarios. To further illustrate this problem, we conduct toy experiments involving clustering tasks on BBCSport and WebKB datasets under different scenarios, including noisy multi-view, clean single-view, and clean multi-view conditions. From the experimental results are presented in Fig. 3 and Tab. 4, it can be observed that when confronted with noisy data, the model's performance degrades drastically, even performing worse than single-view clustering. Therefore, it is critical to mitigate the adverse effects of noisy data in multi-view clustering tasks. In the following sections, we detail our proposed strategies for the automatic identification and rectification of noise.

## 3. Methodology

To mitigate the negative impact of noisy data, we propose a robust Automatic Identification and Rectification deep contrastive Multi-View Clustering framework with noisy view, which termed AIRMVC. The overall framework of AIRMVC is presented in Fig. 2. Specifically, we begin with providing the detailed descriptions of our AIRMVC, including three components, i.e., noisy identification, hybrid rectification, and the noise-robust contrastive mechanism. Then, we define the objective functions to optimize the whole model. Finally, we present the theoretical analysis of our designed noise-robust contrastive component.

### 3.1. Noisy Identification

For a given multi-view dataset $\{x^v\}_{v=1}^V$, we utilize an encoder network to generate the representations, expressed as $\mathbf{E}^v = \mathcal{F}^v(x^v; \Theta^v)$. For each representation, we first process it using a multi-layer perceptron (MLP) layer as the projector. Subsequently, we model its distribution using a Gaussian Mixture Model (GMM). This process can be described as follows:

$$
\begin{aligned}
p(\mathbf{E}) &= \sum_{k=1}^{K} p(\mathbf{E}, q = k) \\
&= \sum_{k=1}^{K} p(q = k) \mathcal{N}(\mathbf{E}|\mu_k, \sigma_k),
\end{aligned}
\tag{1}
$$

where $q \in \{1, 2, \ldots, K\}$ represents the discrete latent variables. $\mu_k$ and $\sigma_k$ denote the mean and variance, respectively. Assuming that $q$ follows a uniform distribution, the probability of $q = k$ could be expressed as $p(q = k) = \frac{1}{K}$. Based on this, the posterior probability of assigning $x_i$ to the $k$-th cluster can be calculated as:

$$
\chi_{ik} = p(q_i = k|x_i) \propto \mathcal{N}(x_i|\mu_k, \sigma_k). \tag{2}
$$

The discrete variables $q$ can correspond directly to the labels in an ideal condition. The entire process can be optimized and solved using the Expectation-Maximization (EM) algorithm. However, in real-world scenarios, noisy

data inputs are pervasive and unavoidable, leading to biases in the optimization process. Thus, the identification and rectification of noisy data are critically important. In unsupervised scenarios, identifying noise remains a challenging task. To address this issue, we establish a connection between the latent variable $q$ and the model prediction $y$ in an unsupervised manner. Specifically, we replace the assignment $p(q_i = k|x_i)$ with the soft prediction of the network $p(y_i = k|x_i)$. The parameters of the GMM can then be computed as follows:

$$
\begin{aligned}
\mu_k &= \text{Norm}\left(\frac{\sum_i p(y_i = k|x_i)\mathbf{E}_i}{\sum_i p(y_i = k|x_i)}\right), \\
\sigma_k &= \frac{\sum_i p(y_i = k|x_i)(\mathbf{E}_i - \mu_k)(\mathbf{E}_i - \mu_k)^\mathsf{T}}{\sum_i p(y_i = k|x_i)},
\end{aligned}
\tag{3}
$$

where Norm means the $\ell_2$ normalization. Based on Eq. (3), we reformulate Eq. (2) by considering the intra-cluster distance. Since we implement $\ell_2$ normalization, we have $(\mathbf{E} - \mu_k)^\mathsf{T}(\mathbf{E} - \mu_k) = 2 - 2\mathbf{E}^\mathsf{T}\mu_k$, the process of Eq. (2) could be expressed by:

$$
\begin{aligned}
\chi_{ik} &= p(q_i = k|x_i) \\
&= \frac{\exp\left(-(\mathbf{E}_i - \mu_k)^\mathsf{T}(\mathbf{E}_i - \mu_k)/2\sigma_k\right)}{\sum_k \exp\left(-(\mathbf{E}_i - \mu_k)^\mathsf{T}(\mathbf{E}_i - \mu_k)/2\sigma_k\right)} \\
&= \frac{\exp\left(\mathbf{E}_i^\mathsf{T}\mu_k/\sigma_k\right)}{\sum_k \exp\left(\mathbf{E}_i^\mathsf{T}\mu_k/\sigma_k\right)}.
\end{aligned}
\tag{4}
$$

In this way, we obtain the soft prediction of a sample belonging to the $k$-th cluster ($\mu_k$). Furthermore, we combine the model's predictions with the representation distribution in an unsupervised manner and update them using the GMM. By formulating the GMM to model the distribution of representations and soft predictions, we transform the noisy identification problem into an anomaly identification problem. In a multi-view setting, if a sample is clean, its soft predictions and cluster assignments should remain consistent across the different views. For a given sample $x_i$, it is classified as either clean (normal data) or noisy (anomalous data). Based on the above analysis, we provide the conditional probability to determine the likelihood of $x_i$ containing clean information, which can be calculated as:

$$
\chi_{y=q|i} = p(y_i = q_i|x_i) = \frac{\exp\left(\mathbf{E}_i^\mathsf{T}\mu_{qi}/\sigma_{qi}\right)}{\sum_k \exp\left(\mathbf{E}_i^\mathsf{T}\mu_k/\sigma_k\right)}.
\tag{5}
$$

After that, we introduce a two-component GMM to automatically identify the clean probability of a given sample, presented as:

$$
p(\chi_{y=q|i}) = \underbrace{p(\chi_{y=q|i}, a = 1)}_{\varphi_i} + \underbrace{p(\chi_{y=q|i}, a = 0)}_{1-\varphi_i}
\tag{6}
$$

where $a = 1$ represents the cluster of clean samples with a higher mean value, while $a = 0$ corresponds to the cluster with a lower mean value. Consequently, $\varphi_i$ can be interpreted as the probability of a sample $x_i$ being clean, whereas $1 - \varphi_i$ denotes the probability of $x_i$ being noisy. Using Eq. (6), we can calculate the posterior probability that determines whether a sample $x_i$ is clean, which is presented in Fig. 2(a).

### 3.2. Hybrid Rectification Strategy Design

In Section 3.1, we estimate the probability $\varphi_i^v$ of each sample being classified as clean in $v$-th view. Following this, we propose a hybrid rectification strategy to address noisy samples. The details of the strategy are illustrated in Fig. 2(b). Specifically, we utilize a combination of predicted soft distributions to perform noise rectification:

$$
\begin{aligned}
y_i^v &= h(\mathbf{E}_i^v), \\
m_i^v &= \varphi_i^v \times y_i^v + (1-\varphi_i^v) \times y_i^1, v = \{2, \ldots, V\},
\end{aligned}
\tag{7}
$$

where $m_i^v$ represents the mixed soft prediction, and $h(\cdot)$ denotes the classifier head. We utilize a multilayer perceptron (MLP) with a softmax function as the backbone for $h(\cdot)$. In this paper, we focus on the unsupervised clustering task, where obtaining reliable supervisory information is particularly challenging. Following previous noise-robust MVC methods (Huang et al., 2020; Yang et al., 2023a; Sun et al., 2024; Yang et al., 2021), we assume the first view to be the clean view. Therefore, in Eq. (6), the mixed soft prediction is predominantly influenced by the clean samples. Conversely, as $\varphi_i$ approaches 0, the mixed soft prediction is adjusted based on the prediction of the first view.

With the mixed soft prediction, we could provide the following rectification loss:

$$
\mathcal{L}_{rs} = \frac{1}{V-1}\sum_{v=2}^{V}\left(-\sum_{i=1}^{N}\sum_{j=1}^{K} m_{ij}^v \log\left(y_{ij}^v\right)\right),
\tag{8}
$$

where $y^v$ represents the $v$-th soft prediction and $v \in \{2, \ldots, V\}$. In this manner, the noisy data could be rectified by the cross-entropy loss. A more detailed experimental analysis of the hybrid rectification strategy is presented in Section 5.3.

### 3.3. Noise-Robust Contrastive Mechanism

Contrastive learning (Yang et al., 2024b) has been proven to be an effective technique for enhancing representation robustness. In this subsection, we propose a noise-robust contrastive mechanism to further mitigate the impact of noisy data in multi-view clustering tasks. In unsupervised multi-view noisy scenarios, constructing positive and negative sample pairs based solely on indices may result in

incorrect pairings (Sun et al., 2024). Ensuring the reliability of sample pair construction is a significant challenge. To improve the accuracy of pair construction, we incorporate soft predictions as an additional validation criterion. Similar to previous methods, the first step is to calculate the similarity between samples across views:

$$s(\mathbf{E}_i^m, \mathbf{E}_j^n) = \frac{\mathrm{sim}(\mathbf{E}_i^m, \mathbf{E}_j^n)}{||\mathbf{E}_i^m||_2 ||\mathbf{E}_j^n||_2}, \tag{9}$$

where $\mathrm{sim}(\cdot)$ denotes the similarity function, e.g., cosine similarity. Next, we present the noise-robust contrastive loss for $i$-th and $j$-th sample in $m$-th and $n$-th view as:

$$\ell^{mn} = \mathbb{I}\{(y_i^m)^\top(y_j^n) \geq \tau\} \tag{10}$$
$$\left(\log(1 - s(\mathbf{E}_i^m, \mathbf{E}_j^n)) + \log(1 - s(\mathbf{E}_j^m, \mathbf{E}_i^n))\right),$$

where $\tau$ is the confidence threshold used to control the construction of sample pairs for contrastive learning by selecting similar samples. As illustrated in Fig.2(c), green denotes correctly constructed sample pairs, while red indicates incorrect ones. Our proposed strategy significantly improves the accuracy of sample pair construction, thereby enhancing the overall reliability of the contrastive learning process. Then, the robust contrastive loss for all cross-view is defined by:

$$\mathcal{L}_{con} = \frac{1}{V(V-1)} \sum_{m=1}^{V} \sum_{n=1}^{V} \ell^{mn}, m \neq n. \tag{11}$$

### 3.4. Objective Function

Our proposed AIRMVC is primarily optimized using three objective functions: reconstruction loss $\mathcal{L}_{rec}$, rectification loss $\mathcal{L}_{rs}$, and robust-noisy contrastive loss $\mathcal{L}_{con}$. To be specific, $\mathcal{L}_{rec}$ denotes the reconstruction procedure with autoencoder network to learn the representations $\mathbf{E}$ in latent space. The process could be formulated as:

$$\mathcal{L}_{rec} = \sum_{v=1}^{V} \sum_{i=1}^{N} ||x_i^v - \mathcal{G}^v(\mathcal{F}^v(x_i^v; \Theta^v); \Phi^v)||_2^2, \tag{12}$$

where $\mathcal{G}^v$ and $\mathcal{F}^v$ are the decoder and encoder network for $v$-th view, respectively. In summary, the overall objective function of AIRMVC is:

$$\mathcal{L} = \mathcal{L}_{rec} + \alpha \cdot \mathcal{L}_{rs} + \beta \cdot \mathcal{L}_{con}, \tag{13}$$

where $\alpha$ and $\beta$ are the trade-off hyper-parameters. By optimizing the overall objective function, AIRMVC could automatically identify and rectify the noisy data in unsupervised multi-view clustering task. Due to the space limited, we present the training algorithm in Alg. 1 in Appendix.

*Table 1.* Statistics summary of six benchmark datasets.

| Dataset | #Views | #Samples | #Clusters |
|---------|--------|----------|-----------|
| BBCSport | 2 | 544 | 5 |
| WebKB | 2 | 1051 | 2 |
| Reuters | 5 | 1200 | 6 |
| UCI-digit | 3 | 2000 | 10 |
| Caltech101 | 5 | 9144 | 102 |
| STL10 | 4 | 13000 | 10 |

## 4. Theoretical Analysis

In this subsection, we examine the rationale behind our proposed noise-robust contrastive mechanism from a theoretical perspective. For clarity, we provide the following definitions of symbols. Let $x$ and $x^+$ denote the input sample and its positive sample in our noise-robust contrastive mechanism. $y$ and $y'$ represent the clean and noisy soft prediction, respectively. We consider $\mathbf{E}^*$ as the representations by maximizing the mutual information between $\mathbf{E}$ and $x+$, i.e., $\mathbf{E}^* = \mathrm{argmax}_{\mathbf{E}} I(\mathbf{E}, x^+)$. Besides, we define $I(x; y|x^+) \leq \vartheta$ and $I(x; y'|x^+) > \eta$. For input sample $x$, the clean soft prediction $y$, and the noisy soft prediction $y'$, we have:

**Theorem 4.1.** *The representations $\mathbf{E}^*$ retain clean information and discard noisy information, which can be presented as:*

$$I(x; y) - \vartheta \leq I(\mathbf{E}^*; y) \leq I(x; y),$$
$$I(\mathbf{E}^*; y') \leq I(x; y') - \eta + \vartheta. \tag{14}$$

**Remark:** For the input samples $x$, the corresponding positive samples $x^+$ and prediction $y_i$, $\vartheta$ could be interpreted as the relatively small information gain contributed by the positive samples $x^+$. For the fixed $x$ and its positive samples $x^+$, the information gain for the class prediction $y$ is limited. In contrast, $x^+$ contributes greater information gain to the noisy prediction $y'$, which we regard as $\eta$. By calculating the mutual information $I(x_i; y_i)$, Theorem. 14 demonstrates that the representations $\mathbf{E}^*$ learned by the contrastive mechanism could discard the noisy information while preserving the clean information. The proof is provided in Section A.2.

## 5. Experiments

In this section, we perform a series of experiments to evaluate the effectiveness and advantages of our proposed method. Specifically, we aim to address the following research questions (**RQs**): **RQ1**: How does AIRMVC compare with other leading deep multi-view clustering techniques in terms of performance? **RQ2**: What art the impacts of the components of AIRMVC to enhance multi-view clustering results? **RQ3**: What clustering structures are identified by AIRMVC? **RQ4**: What is the effect of hyper-parameters on the efficacy of AIRMVC?

*Table 2.* Multi-view clustering performance on six benchmark datasets (Part 1/4). The highest performance metrics are emphasized in red, while the runner-up results are marked in blue.

| | | Noisy Rate | 10% | | | 30% | | | 50% | | | 70% | | |
|---|---|---|---|---|---|---|---|---|---|---|---|---|---|---|
| | | Metric | ACC↑ | NMI↑ | PUR↑ | ACC↑ | NMI↑ | PUR↑ | ACC↑ | NMI↑ | PUR↑ | ACC↑ | NMI↑ | PUR↑ |
| BBCSport | CANDY | NeurIPS 2024 | 23.16 | 05.60 | 40.62 | 19.49 | 03.93 | 38.79 | 17.10 | 03.72 | 38.42 | 16.36 | 02.68 | 36.58 |
| | RMCNC | TKDE 2024 | 35.48 | 15.80 | 35.48 | 30.57 | 12.87 | 30.57 | 28.56 | 10.67 | 28.05 | 24.26 | 04.57 | 24.26 |
| | TGM-MVC | ACM MM 2024 | 37.28 | 08.39 | 40.84 | 36.37 | 06.33 | 38.82 | 30.19 | 05.81 | 30.40 | 25.13 | 04.16 | 25.13 |
| | SCE-MVC | NeurIPS 2024 | 47.06 | 15.96 | 47.98 | 45.40 | 15.67 | 45.96 | 43.57 | 12.72 | 45.22 | 39.73 | 12.95 | 39.46 |
| | MVCAN | CVPR 2024 | 44.85 | 18.99 | 44.85 | 39.15 | 15.81 | 39.15 | 26.65 | 11.85 | 25.97 | 24.26 | 06.57 | 23.76 |
| | DIVIDE | AAAI 2024 | 31.25 | 04.57 | 38.60 | 30.15 | 03.28 | 36.76 | 28.68 | 02.65 | 35.66 | 27.02 | 02.22 | 36.03 |
| | AIRMVC | Ours | 49.45 | 28.63 | 49.44 | 45.59 | 20.11 | 47.24 | 45.04 | 23.34 | 53.13 | 40.99 | 16.52 | 45.59 |
| WebKB | CANDY | NeurIPS 2024 | 19.03 | 12.41 | 19.03 | 18.08 | 10.85 | 18.10 | 16.56 | 07.84 | 16.56 | 15.60 | 04.87 | 15.60 |
| | RMCNC | TKDE 2024 | 78.12 | 10.77 | 78.10 | 66.98 | 08.78 | 66.98 | 60.04 | 06.24 | 60.00 | 52.71 | 04.35 | 52.16 |
| | TGM-MVC | ACM MM 2024 | 76.86 | 13.81 | 76.86 | 71.87 | 11.65 | 71.87 | 62.86 | 08.04 | 62.86 | 55.51 | 06.87 | 55.50 |
| | MVCAN | CVPR 2024 | 77.83 | 12.39 | 78.12 | 67.46 | 06.20 | 67.46 | 64.22 | 05.94 | 64.22 | 55.66 | 05.74 | 55.65 |
| | SCE-MVC | NeurIPS 2024 | 76.78 | 18.78 | 69.65 | 74.12 | 13.22 | 68.19 | 65.75 | 08.17 | 62.18 | 60.08 | 03.52 | 60.77 |
| | DIVIDE | AAAI 2024 | 66.41 | 15.14 | 69.28 | 53.66 | 10.91 | 57.65 | 52.24 | 07.70 | 56.26 | 51.38 | 05.60 | 55.34 |
| | AIRMVC | Ours | 83.73 | 27.15 | 83.73 | 77.93 | 13.48 | 78.12 | 74.98 | 08.42 | 74.98 | 62.89 | 06.69 | 62.32 |
| Reuters | CANDY | NeurIPS 2024 | 24.58 | 08.50 | 33.75 | 22.17 | 08.09 | 33.08 | 20.83 | 05.57 | 29.25 | 18.83 | 04.44 | 26.75 |
| | RMCNC | TKDE 2024 | 39.42 | 24.21 | 41.83 | 35.83 | 16.96 | 37.83 | 30.17 | 10.92 | 31.67 | 26.83 | 09.07 | 28.75 |
| | TGM-MVC | ACM MM 2024 | 31.28 | 09.05 | 31.53 | 30.30 | 08.51 | 30.30 | 26.44 | 06.78 | 27.59 | 23.90 | 04.58 | 25.42 |
| | SCE-MVC | NeurIPS 2024 | 43.25 | 22.89 | 43.23 | 41.25 | 20.00 | 41.92 | 39.92 | 22.42 | 39.09 | 39.33 | 19.87 | 39.08 |
| | MVCAN | CVPR 2024 | 45.25 | 23.50 | 45.50 | 37.67 | 14.53 | 38.75 | 29.67 | 14.04 | 30.25 | 23.25 | 18.47 | 25.08 |
| | DIVIDE | AAAI 2024 | 41.08 | 16.82 | 42.50 | 33.42 | 13.42 | 34.42 | 30.50 | 10.03 | 33.25 | 27.92 | 09.39 | 28.61 |
| | AIRMVC | Ours | 48.25 | 26.34 | 48.25 | 46.67 | 23.54 | 46.67 | 44.83 | 21.72 | 45.58 | 41.75 | 20.91 | 42.33 |

## 5.1. Experimental Settings

**Datasets:** To evaluate the effectiveness of the proposed AIRMVC framework, we implement comprehensive experiments on six widely used datasets: BBCSport, Reuters, Caltech101, UCI-digit, WebKB, SUNRGB-D, and STL10. Tab. 1 provides detailed statistical information for all datasets. To simulate a noisy input environment, we randomly introduce noise into the multi-view input data $x$ at varying proportions of {10%, 30%, 50%, 70%, 90%}.

**Evaluation Metrics:** To provide a thorough evaluation of the model's clustering performance, we utilize three widely recognized metrics: Accuracy (ACC), Normalized Mutual Information (NMI), and Purity (PUR). To ensure a fair comparison, all methods are assessed across 10 independent runs, and the average results are reported.

**Comparison algorithms:** To showcase the broad applicability and superior performance of the proposed AIRMVC, we compare it against 11 state-of-the-art deep multi-view clustering methods. These baselines are categorized into two groups: classical deep multi-view clustering methods (CoMVC (Trosten et al., 2021), SiMVC (Trosten et al., 2021), MFLVC (Xu et al., 2022b), DealMVC (Yang et al., 2023b), SURE (Yang et al., 2023a), CANDY (Guo et al.), DIVIDE (Lu et al., 2024), TGM-MVC (Wang et al., 2024a), and SCE-MVC (Wang et al., 2024b)) and noise-resilient deep multi-view clustering methods (RMCNC (Sun et al., 2024) and MVCAN (Xu et al., 2024)). Detailed information on these baselines is provided in Section. A.5 of the Appendix.

**Implement Details:** In this study, all experiments are con-

ducted on the PyTorch (Imambi et al., 2021) platform using an NVIDIA A6000 GPU. For fair comparison, we reproduce the results of all baselines using the original source codes and configurations provided by their authors. Our proposed AIRMVC is trained using the Adam optimizer (Kingma & Ba, 2014) with its default settings. To enhance the discriminative power of the network and obtain reliable soft predictions in an unsupervised setting, we pre-train the model with an auto-encoder module for 100 epochs. The trade-off hyperparameters $\alpha$ and $\beta$ are consistently set to 1.0. The maximal training epoch is set to 400 for all datasets. A detailed overview of the hyperparameter settings is provided in Tab. 7 in the Appendix.

## 5.2. Comparison Experiments (RQ1)

In this subsection, we conduct comprehensive experiments to demonstrate the superior and effectiveness of our designed AIRMVC. To be specific, we compare with 11 state-of-the-art baselines with different noise ratio. The experimental results are presented in Tab. 2, 3, 6, 8, 9. We highlight the optimal results in bold red, and the sub-optimal results with blue. Due to the space limited, we present Tab. 6, 8, 9 in Section. A.3 of Appendix. From the results, we could observe the following observations:

1) AIRMVC significantly outperforms state-of-the-art baselines across most metrics and datasets. To be specific, compared to the best multi-view clustering algorithm, AIRMVC achieves remarkable improvements by 2.39%, 9.64%, and 1.46% w.r.t. ACC, NMI and PUR in BBCSport dataset with 10% noise. We analyze the reason is that AIRMVC could

*Table 3.* Multi-view clustering performance on six benchmark datasets (Part 2/4). The highest performance metrics are emphasized in red, while the runner-up results are marked in blue.

| | Noisy Rate | | 10% | | | 30% | | | 50% | | | 70% | | |
|---|---|---|---|---|---|---|---|---|---|---|---|---|---|---|
| | Metric | | ACC↑ | NMI↑ | PUR↑ | ACC↑ | NMI↑ | PUR↑ | ACC↑ | NMI↑ | PUR↑ | ACC↑ | NMI↑ | PUR↑ |
| **UCI-digit** | CANDY | NeurIPS 2024 | 80.90 | 67.70 | 80.90 | 72.30 | 61.19 | 72.30 | 67.10 | 54.94 | 67.10 | 63.10 | 52.50 | 65.20 |
| | RMCNC | TKDE 2024 | 27.40 | 14.74 | 29.50 | 25.20 | 12.17 | 26.80 | 23.65 | 11.22 | 25.30 | 22.20 | 10.10 | 24.05 |
| | TGM-MVC | ACM MM 2024 | 60.23 | 65.08 | 64.89 | 59.52 | 64.03 | 63.60 | 56.31 | 60.23 | 60.25 | 52.01 | 56.88 | 58.28 |
| | SCE-MVC | NeurIPS 2024 | 72.60 | 71.70 | 72.55 | 68.30 | 67.49 | 70.00 | 66.55 | 67.10 | 68.80 | 64.45 | 55.58 | 66.30 |
| | MVCAN | CVPR 2024 | 86.50 | 79.00 | 86.50 | 74.35 | 72.78 | 76.90 | 64.15 | 66.54 | 67.65 | 63.10 | 54.21 | 63.24 |
| | DIVIDE | AAAI 2024 | 88.20 | 80.28 | 88.20 | 76.00 | 71.56 | 78.45 | 71.00 | 62.80 | 74.20 | 66.25 | 54.72 | 68.25 |
| | AIRMVC | Ours | 93.95 | 89.08 | 93.95 | 77.60 | 76.65 | 80.75 | 76.05 | 68.17 | 76.05 | 69.20 | 60.38 | 69.20 |
| **Caltech101** | CANDY | NeurIPS 2024 | 16.67 | 16.93 | 19.67 | 15.87 | 16.62 | 19.19 | 10.64 | 10.91 | 10.94 | 10.98 | 15.14 | 17.35 |
| | RMCNC | TKDE 2024 | 08.75 | 16.57 | 08.71 | 07.20 | 16.02 | 07.21 | 06.94 | 12.87 | 06.87 | 06.71 | 11.58 | 06.71 |
| | TGM-MVC | ACM MM 2024 | 12.81 | 31.50 | 28.95 | 10.58 | 26.57 | 26.25 | 07.16 | 21.84 | 19.54 | 06.66 | 20.46 | 18.57 |
| | MVCAN | CVPR 2024 | 20.53 | 35.81 | 24.60 | 14.30 | 26.02 | 14.14 | 10.28 | 23.19 | 10.99 | 10.34 | 20.43 | 10.44 |
| | SCE-MVC | NeurIPS 2024 | 20.33 | 35.28 | 17.62 | 14.25 | 25.74 | 14.01 | 10.78 | 20.51 | 10.44 | 10.77 | 19.46 | 08.54 |
| | DIVIDE | AAAI 2024 | 18.03 | 35.40 | 20.41 | 15.10 | 25.80 | 20.08 | 11.10 | 23.33 | 16.33 | 10.16 | 18.88 | 14.87 |
| | AIRMVC | Ours | 21.45 | 37.16 | 34.69 | 15.89 | 29.39 | 27.60 | 11.80 | 23.66 | 23.33 | 11.89 | 21.14 | 20.14 |
| **STL10** | CANDY | NeurIPS 2024 | 27.08 | 21.03 | 27.33 | 20.56 | 19.23 | 20.21 | 18.19 | 18.48 | 18.77 | 17.49 | 16.79 | 17.80 |
| | RMCNC | TKDE 2024 | 16.51 | 03.56 | 16.92 | 15.85 | 02.94 | 16.22 | 14.78 | 02.33 | 15.12 | 14.12 | 01.70 | 14.95 |
| | TGM-MVC | ACM MM 2024 | 27.50 | 19.51 | 28.10 | 21.80 | 21.62 | 21.65 | 20.01 | 20.02 | 20.78 | 19.87 | 14.05 | 19.77 |
| | SCE-MVC | NeurIPS 2024 | 27.58 | 23.97 | 27.05 | 20.77 | 19.34 | 20.92 | 18.45 | 14.51 | 20.10 | 15.85 | 12.60 | 16.93 |
| | MVCAN | CVPR 2024 | 26.64 | 24.05 | 26.57 | 21.62 | 20.17 | 20.52 | 19.36 | 20.01 | 19.09 | 17.25 | 17.96 | 17.37 |
| | DIVIDE | AAAI 2024 | 27.58 | 24.54 | 27.08 | 22.03 | 20.55 | 22.17 | 20.78 | 20.25 | 20.61 | 16.86 | 20.25 | 20.28 |
| | AIRMVC | Ours | 28.81 | 25.04 | 29.01 | 22.46 | 23.64 | 22.46 | 21.95 | 23.09 | 22.02 | 20.14 | 22.66 | 20.14 |

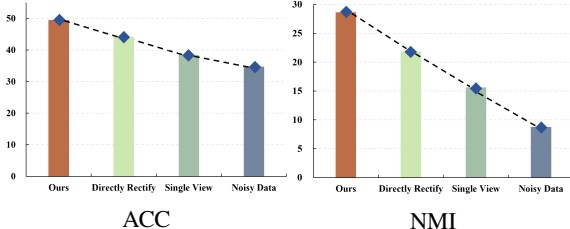

ACC                NMI

*Figure 3.* Ablation studies for our proposed noisy-identification and rectification strategy on BBCSport dataset.

automatically identify and rectify the noisy data, leading to a robust learning procedure.

2) When noise is present in multi-view data, the performance of deep multi-view clustering models tends to become unstable. The clustering performance of the model exhibits a downward trend as the noise proportion increases. We attribute this decline to the distortion of the underlying clustering structure caused by the presence of noisy data.

3) Compared with the classical deep multi-view clustering methods, e.g., TGM-MVC (Wang et al., 2024a), our AIR-MVC could achieve better performance. These methods lack specifically designed strategies to mitigate the adverse effects of noise, which consequently leads to a decline in performance.

4) Although noise deep multi-view clustering methods could obtain promising performance, e.g., MVCAN (Xu et al., 2024), the impact of noise has not been fully eliminated in complex noisy scenario. The noisy identification and rectification strategies and noise-robust contrastive mechanism in AIRMVC could identify and rectify the noisy data. Thus, achieving better performance.

*Table 4.* Performance variation of multi-view clustering on WebKB dataset in noisy scenario with 10% noisy ratio.

| Methods | TGM-MVC | | | SCE-MVC | | |
|---|---|---|---|---|---|---|
| Metrics | ACC | NMI | PUR | ACC | NMI | PUR |
| Clean | 81.45 | 15.70 | 81.45 | 83.43 | 20.64 | 83.43 |
| Noise | 76.86↓ | 13.81↓ | 76.86↓ | 76.78↓ | 18.78↓ | 69.65↓ |

In summary, AIRMVC enables robust training under varying levels of noise. Moreover, the above observations and experimental results further validate the effectiveness and generalization capability of AIRMVC.

**5.3. Ablation Study (RQ2)**

In this subsection, we conduct experiments to investigate the effectiveness of the components in AIRMVC. Due to the space limited, we present the experimental results with 30% noise ratio in Section. A.3.2 of Appendix.

**Effectiveness of design modules:** Here, we implement ablation studies to verify the designed modules, including the noisy identification and rectification strategy and noise-robust contrastive mechanism. In this subsection, we denote "(w/o) D&R", "(w/o) Con", and "(w/o) D&R&Con" represent the model to remove noisy identification and rectification strategy, noise-robust contrastive mechanism, and both modules combined, respectively. Besides, we leverage an autoencoder network as the backbone for "(w/o) D&R&Con" to derive representations for the downstream clustering task. We conduct experiments with three metrics on six datasets. The experimental results are demonstrated in Fig. 4. From those results, we could conclude the following observations:

1) When any component of our designed model is removed, the performance of the model degrades. This indicates that

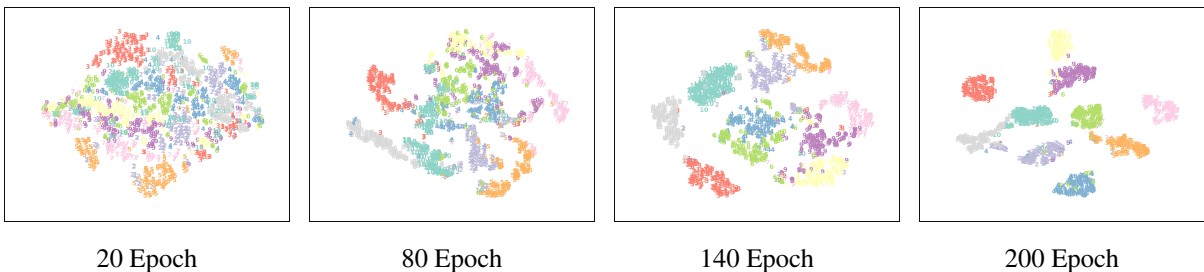

*Figure 4.* Ablation studies on BBCSport, Caltech101, STL10, and Reuters datasets with 10% noisy ratio.

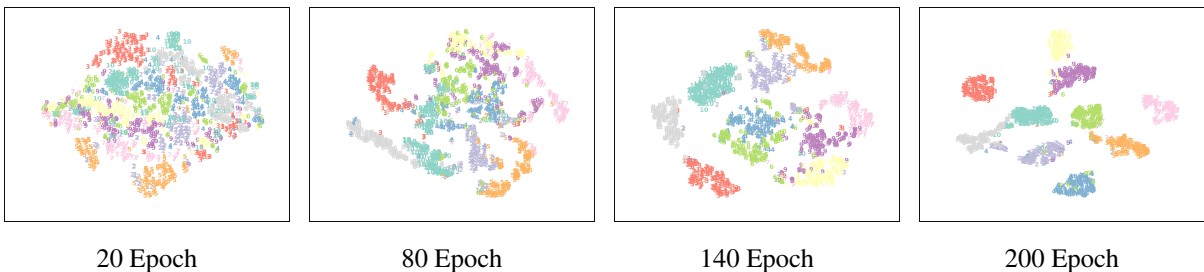

20 Epoch        80 Epoch        140 Epoch        200 Epoch

*Figure 5.* Visualization of the representations during the training process on UCI-digit dataset.

each part of our design contributes to the overall clustering performance.

2) The removal of noisy identification and rectification strategy ("(w/o) D&R") results in a more substantial performance degradation compared to the removal of noise-robust contrastive mechanism ("(w/o) Con"), highlighting the relative importance of noisy identification and rectification strategy to the model's performance.

**Effectiveness of rectification strategy:**

In AIRMVC, we rectify the noisy data with two steps. Specifically, we first employ two-component GMM to identify the noisy data. Then, we rectify the noisy data based on identification result. To illustrate the adverse effects of noise, we present experimental evidence in Tab. 4 using the WebKB dataset, demonstrating a significant performance degradation in noisy scenarios. This underscores the necessity of designing a strategy specifically targeted at noise identification and rectification. To further validate the effectiveness of our noise identification and rectification strategy, we conduct experiments on the BBCSport dataset with a 10% noise ratio. For a comprehensive evaluation, we define four different experimental setups. For simplicity, we use "Noisy Data" to represent the direct fusion of noisy multi-view features, "Single View" to represent the results from a single noisy view, "Directly Rectify" to represent directly using cross-entropy with the first view to rectify, and "Ours" to represent our proposed strategy. The experimental results are presented in Fig. 3. From these results, we can derive the following observations:

1) Directly fusing noisy multi-view data results in the worst clustering performance. We attribute this phenomenon to the lack of any noisy rectification mechanism, allowing

the noisy views to mutually amplify their adverse effects. Furthermore, the fusion process exacerbates the impact of noise. Without rectification, the clustering performance of multi-view data is even worse than that of a single view.

2) Compared to the directly rectification with the first view, our proposed AIRMVC could achieve better performance. This phenomenon can be attributed to the tendency of directly using the correction results from the first view to homogenize features across different views. In contrast, our noisy identification and rectification strategy effectively filters out noisy data from each view while retaining complementary information that is beneficial for downstream tasks.

### 5.4. Visualization Analysis (RQ3)

To visualize the latent representation space and assess the effectiveness of AIRMVC, we employ $t$-SNE (Van der Maaten & Hinton, 2008) as the visualization tool in our experiments. Specifically, we extract representations from the UCI-digit dataset. The results, as shown in Fig. 5, reveal the following observations. As training progresses, the cluster structures within the UCI-digit dataset become increasingly distinct. By the 200th epoch, well-defined and distinguishable cluster structures have emerged. This observation demonstrates the capability of AIRMVC to effectively uncover the latent cluster structures within feature representations.

### 5.5. Hyper-parameter Analysis (RQ4)

In this subsection, we conduct experiments to analysis the sensitivity of the hyper-parameters in this paper.

**Sensitive analysis of trade-off hyper-parameter $\alpha$ and $\beta$:**

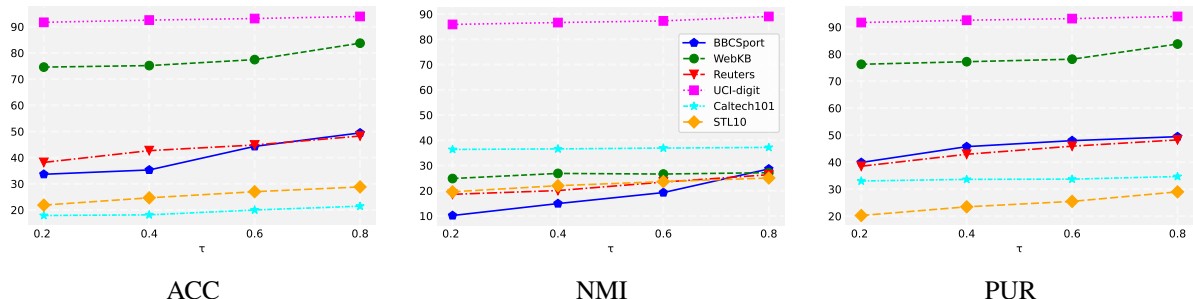

| ACC | NMI | PUR |

*Figure 6.* Sensitive analysis on hyper-parameter $\tau$ on six datasets with 10% noisy ratio.

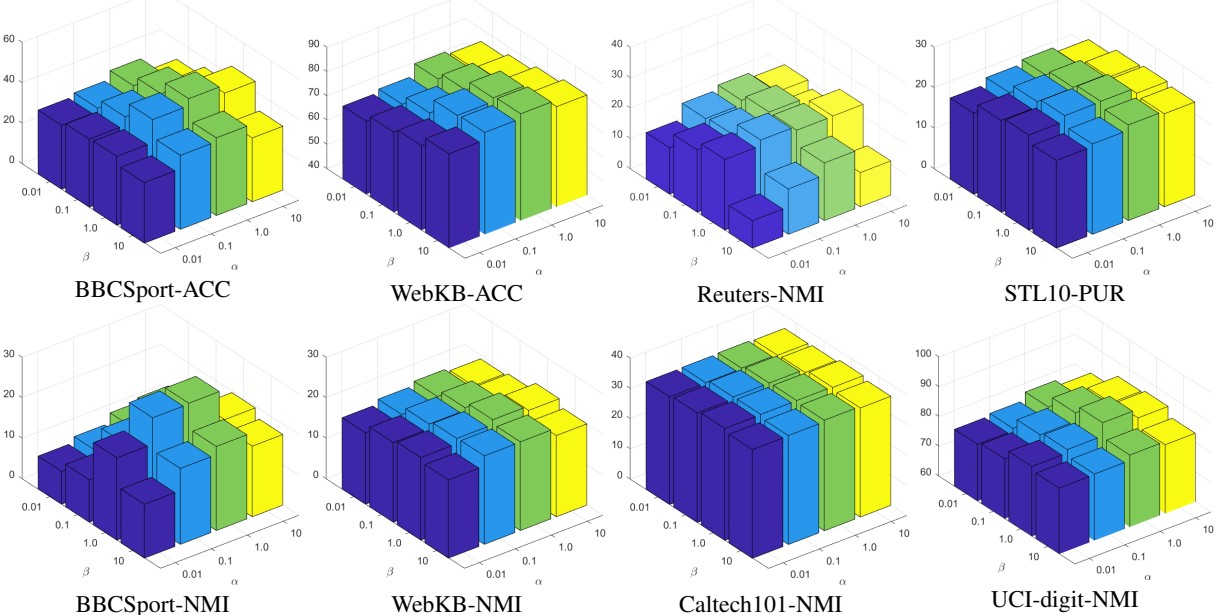

*Figure 7.* Sensitivity Analysis for $\alpha$ and $\beta$ with ACC, NMI, and PUR on BBCSport, WebKB, Reuters, UCI-digits, Caltech101 and STL10 datasets with 10% noisy ratio.

To further examine the impact of the parameters $\alpha$ and $\beta$ on our model, we performed sensitive experiments on six datasets with 10% noise ratio. Specifically, we analyze parameter values within the range of $\{0.01, 0.1, 1.0, 10\}$. According to the results presented in Fig. 7. We could find the following observations. 1) When $\alpha$ and $\beta$ approach extreme values, the model's clustering performance deteriorates significantly. This decline can be attributed to the disruption of the balance in the loss function. We observe that the model achieves optimal clustering performance when the values of $\alpha$ and $\beta$ are set to 1.0. Thus, we set the parameters to 1.0. 2) Compared to changes in $\beta$, variations in $\alpha$ result in more pronounced changes in the model's performance. This indicates that the noisy identification and rectification module contributes more significantly to enhancing performance.

**Sensitive analysis of threshold $\tau$:**

We conduct experiments to evaluate the influence of the threshold parameter $\tau$. We varied the value of $\tau$ within the range of $\{0.2, 0.4, 0.6, 0.8\}$. The results are demonstrated in

Fig. 6. We have the following observation. As the threshold increases, the clustering performance of the model improves progressively. We attribute this to the fact that higher thresholds reduce the number of incorrect contrastive learning sample pairs.

## 6. Conclusion

In this work, we present a novel deep contrastive multi-view clustering network to automatically identify and rectify the noisy data (AIRMVC). In AIRMVC, we first consider the noisy identification as an anomaly identification problem. Then, based on the identification results, we introduce a hybrid rectification strategy to alleviate the adverse impact of noisy data. After that, we design a noise-robust contrastive mechanism to obtain more discriminative representations. Moreover, we theoretical proof that the learned representations could discard the noisy information. Comprehensive experiments conducted on six benchmark datasets validate the effectiveness and robustness of AIRMVC.

## Acknowledgments

This work was supported by the National Science and Technology Innovation 2030 Major Project under Grant No. 2022ZD0209103, the National Natural Science Foundation of China (project No. 62325604, 62441618, 62476281, U24A20323 and 62276271), and the Program of China Scholarship Council (No. 202406110009).

## Impact Statement

This paper presents work whose goal is to advance the field of Machine Learning. There are many potential societal consequences of our work, none of which we feel must be specifically highlighted here.

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

# A. Appendix

## A.1. Notations & Related Work

### A.1.1. NOTATIONS

In this subsection, we provide the notations in Tab. 5.

Table 5. Notation Summary in AIRMVC.

| Notations | Meaning |
|---|---|
| $V$ | The number of view |
| $N$ | The number of samples in each view |
| $K$ | The number of class |
| $x^v$ | The $v$-th input data from multi-view |
| $\mathbf{E}$ | The extracted representations |
| $y$ | The soft prediction of class probability |
| $y'$ | The noisy soft prediction of class probability |
| $\mathcal{F}^v$ | The encoder network |
| $\mathcal{G}^v$ | The decoder network |
| $\Theta^v$ | The parameters of encoder network |
| $\Phi^v$ | The parameters of decoder network |
| $m$ | The mixed soft prediction of class probability |
| $h(\cdot)$ | The classifier head |
| $\varphi$ | The clean probability |
| $s(;)$ | Similarity function |
| $\alpha, \beta$ | The trade-off hyper-parameters |
| $\mathcal{L}_{rs}$ | The rectification loss |
| $\mathcal{L}_{con}$ | The contrastive loss |
| $\mathcal{L}_{rec}$ | The reconstruction loss |

### A.1.2. RELATED WORK

**Contrastive Learning:** Contrastive learning (Yang et al., 2022), recognized for its ability to extract robust intrinsic supervisory signals, has attracted considerable attention in various fields, e.g., recommender system (Yang et al., 2025a;b; Zhao et al., 2023a;b; Dang et al., 2025b;a; Yin et al., 2023; 2024; Yang et al., 2024a), graph learning (Liu et al., 2022b; 2023; Yang et al., 2023c; 2024c; 2023d; Yu et al., 2025; 2024a; Mo et al., 2025; 2024; Huang et al., 2024b;a). The core concept of contrastive learning focuses on maximizing the similarity between positive samples while minimizing the similarity between negative samples within the latent space. Noise Contrastive Estimation (NCE) (Hinton, 2002; Hyvärinen & Dayan, 2005) was introduced, followed by InfoNCE (Van den Oord et al., 2018), which aimed at distinguishing different views of a sample in the presence of others. Subsequently, methods like MoCo (He et al., 2020) and SimCLR (Chen et al., 2020) have advanced the learning of image-specific features by bringing positive pairs closer and pushing negative pairs further apart. In the realm of multi-view clustering, several contrastive learning approaches have been proposed (Tian et al., 2020; Hassani & Khasahmadi, 2020; Xu et al., 2022b). For instance, CMC (Tian et al., 2020) introduced a contrastive multi-view coding framework to extract meaningful semantic information. MVGRL (Hassani & Khasahmadi, 2020) utilized a graph diffusion matrix to generate augmented graphs, which were then leveraged to establish a multi-view contrastive mechanism for downstream tasks.

More recently, MFLVC (Xu et al., 2022b) proposed two objectives for multi-view clustering using high-level features and pseudo-labels within a contrastive learning framework. DealMVC (Yang et al., 2023b) introduced a dual-calibration mechanism specifically designed for deep multi-view clustering, with a calibration contrastive loss that effectively differentiates similar yet distinct samples across multiple views. To further enhance the quality of contrastive learning sample pairs, DIVIDE (Lu et al., 2024) adopted a random walk approach to iteratively identify reliable data pairs, thus improving the stability and robustness of contrastive learning in multi-view clustering. CANDY (Guo et al.) exploits inter-view similarities as contextual cues to uncover false negatives, while also integrating a spectral-based denoising module to refine correspondence in multi-view settings. Contrastive learning, as a powerful method for improving feature quality, has proven

highly effective in the MVC field. In this work, we propose a noise-robust contrastive learning mechanism to enhance the model's performance and robustness in noisy environments.

**Multi-view Learning:** Recently, Multi-view Clustering (MVC) (Wan et al., 2022; 2024; Dong et al., 2023b;a; Yu et al., 2023a; 2024b; 2023b; Li et al., 2025; 2023; Zheng et al., 2024) has attracted substantial research interest. Existing MVC approaches can generally be divided into two categories based on cross-view correspondence: MVC with fully aligned data and MVC with partially aligned data. Fully aligned data assumes predefined mapping relationships between every pair of cross-view samples. Several methods have been developed under this assumption, which can be broadly grouped into five main categories: (1) Non-negative matrix factorization-based methods (Wen et al., 2018), which aim to discover a shared latent factor to integrate information from multiple views; (2) Kernel learning-based methods (Liu et al., 2021), where a predefined set of base kernels is assigned to each view, and the kernel weights are optimally fused to improve clustering performance; (3) Subspace-based methods (Liu et al., 2022a), which assume that all views share a common low-dimensional latent space and derive clustering results by learning this shared representation; (4) Graph-based methods (Li et al., 2021), which construct a unified graph from multi-view data and utilize spectral decomposition to obtain clustering outcomes; and (5) Deep learning-based methods (Yang et al., 2023b), which leverage the powerful representation capabilities of deep neural networks to capture more complex and robust features for clustering.

While these methods have shown strong clustering performance, they typically assume that the features of all views are noise-free. In real-world scenarios, however, noise is often present and can significantly degrade their effectiveness. To address the challenge of noisy data, MVCAN (Xu et al., 2024) adopts an unshared network structure and introduces a two-level optimization strategy to enhance robustness in noisy settings. Similarly, RMCNC (Sun et al., 2024) incorporates a noise-tolerant contrastive loss to counteract the effects of noise in multi-view data. Despite these advancements, existing methods still lack dedicated strategies for explicitly detecting and correcting noise.

To bridge this gap, this paper introduces a novel multi-view clustering framework designed to automatically identify and rectify noise. This framework aims to enhance clustering performance and robustness in the presence of real-world noisy data.

### A.2. Proof of Theorem. 14

In this subsection, we provide the detailed proof of Theorem. 14.

**Theorem A.1.** *The representations $\boldsymbol{E}^*$ retain clean information and discard noisy information, which can be presented as:*

$$I(x;y) - \vartheta \leq I(\boldsymbol{E}^*;y) \leq I(x;y),$$
$$I(\boldsymbol{E}^*;y') \leq I(x;y') - \eta + \vartheta.$$

*Proof.* To proof Theorem. 14, we first present the following definitions.

**Definition A.2** (Mutual Information). The mutual information for the representations and input data in multi-view scenario could be expressed as:

$$I(\mathbf{E};x^+) = \int \int p(\mathbf{E},x^+)\log\frac{p(\mathbf{E}|x^+)}{p(\mathbf{E})}dx^+d\mathbf{E} \tag{15}$$

where $x^+$ is the positive samples for $x$. $\mathbf{E}$ is the representations extracted by the encoder network in latent space.

**Definition A.3** (Relationship for Mutual Information and Representation). The learned representations $\mathbf{E}$ by minimizing Eq. (10) maximize the mutual information, i.e., $I(\mathbf{E};x^+)$.

In detail, Eq. (10) maximizes the similarity of the positive samples while minimizing the similarity of negative samples. The estimate of $\mathbf{E}$ conditioned on $x^+$ is more accurate since $x^+$ is the similar with $x$. Therefore, the mutual information $\log\frac{p(\mathbf{E}|x^+)}{p(\mathbf{E})}$ will increase by minimizing Eq. (10). The similar definition is widely adopted in many other fields (Tsai et al., 2020; Hjelm et al., 2018). Based on Definition. A.3, we define $\mathbf{E}^* = \text{argmax}_{\mathbf{E}}I(\mathbf{E};x^+)$ to denote the representations by minimizing the contrastive loss, i.e., maximizing the mutual information.

**Definition A.4** (Mutual Information Constraint). For the input data $x$, positive samples $x^+$, the clean prediction $y$, and noisy prediction $y'$, we define the mutual information constraint if exists $I(x;y|x^+) \leq \vartheta$ and $I(x;y'|x^+) > \eta$.

Specifically, For the fixed $x$ and its positive samples $x^+$, the information gain for the class prediction $y$ is limited. $\vartheta$ could be interpreted as a small value of the information gain contributed by positive samples $x^+$. In contrast, $x^+$ contributes greater information gain to the noisy prediction $y'$, which we regard as $\eta$.

Based on the above definitions, we present detailed proof. Similar to (Tsai et al., 2020), the first step is to proof the first inequality. With adopting the Data Processing Inequality (Cover, 1999) in the Markov chain $y \leftrightarrow x \rightarrow \mathbf{E}$, for any $\mathbf{E}$ we have:

$$I(x; y) \leq I(\mathbf{E}; y) \tag{16}$$

Therefore, we conclude $I(x; y) \leq I(\mathbf{E}^*; y)$. Moreover, the learned representations $\mathbf{E}^*$ maximize $I(\mathbf{E}; x^+)$ and $I(\mathbf{E}^*; x^+)$ is maximized at $I(x; x^+)$, we could obtain:

$$
\begin{aligned}
I(\mathbf{E}^*; x^+) &= I(x; x^+), \\
I(\mathbf{E}^*; x^*|y) &= I(x; x^+|y)
\end{aligned} \tag{17}
$$

Thus, for distribution $(\mathbf{E}^*, x^+, y)$, we have $I(\mathbf{E}^*; x^+; y) = I(\mathbf{E}^*; x^+) - I(\mathbf{E}^*; x^+|y)$. By substitution, we could obtain:

$$
\begin{aligned}
I(\mathbf{E}^*; x^+; y) &= I(x; x^+) - I(x; x^+|y) \\
&= I(x; x^+; y)
\end{aligned} \tag{18}
$$

After that, we have:

$$
\begin{aligned}
I(\mathbf{E}^*; y) &= I(x; x^+; y) + I(\mathbf{E}^*; y|x^+) \\
&= I(x; y) - I(x; y|x^+) + I(\mathbf{E}^*; y|x^+)
\end{aligned} \tag{19}
$$

Therefore, we have:

$$
\begin{aligned}
I(\mathbf{E}^*; y) &\leq I(x; y), \\
I(\mathbf{E}^*; y) &\geq I(x; y) - I(x; y|x^+) \geq I(x; y) - \vartheta
\end{aligned} \tag{20}
$$

For the second inequality, we could obtain the following formula by expanding $I(\mathbf{E}*, y')$:

$$I(\mathbf{E}^*; y') = I(x; y') - I(x; y'|x^+) + I(\mathbf{E}^*; y'|x^+) \tag{21}$$

Moreover, with the $\eta$ in Definition. A.4 and the Markov Chain $y' \leftarrow y \leftrightarrow x \rightarrow \mathbf{E}$, we have:

$$
\begin{aligned}
I(\mathbf{E}^*; y') &\leq I(x; y') - \eta + I(\mathbf{E}^*; y'|x^+) \\
&I(x; y') - \eta + I(\mathbf{E}^*; y|x^+) \\
&I(x; y') - \eta + \vartheta
\end{aligned} \tag{22}
$$

Therefore, we have completed the proof.

$\square$

## A.3. Additional Experimental Results

In this section, due to the limitation of the original paper pages, we conduct additional experiments including comparison experiments, sensitive analysis, and visualization analysis experiments.

### A.3.1. COMPARISON EXPERIMENTS

We employ experiments with five deep multi-view clustering methods, i.e., SiMVC (Trosten et al., 2021), CoMVC (Trosten et al., 2021), MFLVC (Xu et al., 2022b), DealMVC (Yang et al., 2023b), SURE (Yang et al., 2023a). The results are presented in Tab. 6, 8, 9. Especially, Tab. 9 presents the experimental results with 90% noisy ratio. From those experimental results, we could conclude the following observations.

- AIRMVC consistently outperforms state-of-the-art baselines across various metrics and datasets. Specifically, on the BBCSport dataset with 10% noise, AIRMVC achieves notable improvements of 2.58%, 9.94%, and 2.56% in ACC, NMI, and PUR, respectively, compared to the best multi-view clustering algorithm. This improvement can be attributed to AIRMVC's ability to automatically identify and rectify noisy data, ensuring a more robust learning process.

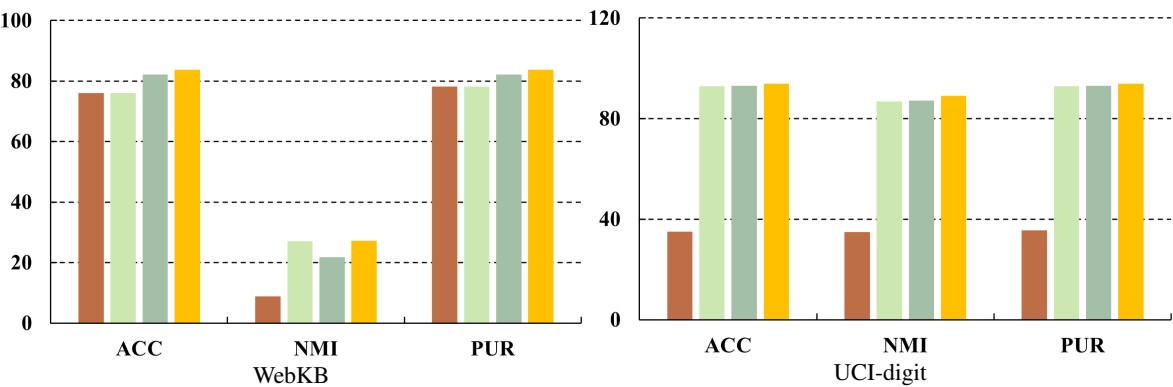

*Figure 8.* Ablation studies on UCI-digit and WebKB datasets with 10% noisy ratio.

- The presence of noise in multi-view data often destabilizes the performance of deep multi-view clustering models. As the proportion of noise increases, the clustering performance deteriorates. This decline can be explained by the distortion of the underlying clustering structures caused by noisy data.

### A.3.2. ABLATION STUDY

In this subsection, we conduct experiments to evaluate the effectiveness of the components within AIRMVC. Due to space constraints, we present the experimental results under a 30% noise ratio.

We perform ablation studies to validate the contributions of the designed components, including the noisy identification and rectification strategy and the noise-robust contrastive mechanism. Specifically, we use the following notations for the ablation models: "(w/o) D&R" refers to removing the noisy identification and rectification strategy, "(w/o) Con" denotes removing the noise-robust contrastive mechanism, and "(w/o) D&R&Con" indicates the removal of both modules. Additionally, in "(w/o) D&R&Con," an autoencoder network is utilized as the backbone to extract representations for downstream clustering tasks. We conduct experiments using three metrics across six datasets, with the results shown in Fig. 9. Based on the results, we draw the following observations:

- Removing any component from the proposed model results in noticeable performance degradation, demonstrating that each module contributes to the overall clustering effectiveness.

- The removal of the noisy identification and rectification strategy ("(w/o) D&R") causes a greater decline in performance than removing the noise-robust contrastive mechanism ("(w/o) Con"), underscoring the critical importance of noise identification and rectification for maintaining robust clustering performance.

### A.3.3. DETAILED HYPER-PARAMETER ANALYSIS

In this subsection, we conduct detailed experiments to analysis the sensitivity of the hyper-parameter $\alpha$ and $\beta$ in this paper.

**Sensitive analysis of trade-off hyper-parameter $\alpha$ and $\beta$:**

To further examine the impact of the parameters $\alpha$ and $\beta$ on our model, we performed sensitive experiments on six datasets with 10% noise ratio. In particular, we analyze parameter values within the range of $\{0.01, 0.1, 1.0, 10\}$. According to the results presented in Fig. 10. We could find the following observations.

- When $\alpha$ and $\beta$ approach extreme values, e.g., $0.01$, the model's clustering performance deteriorates significantly. This decline can be attributed to the disruption of the balance in the loss function. Besides, we could observe that the model achieves optimal clustering performance when the values of $\alpha$ and $\beta$ are set to $1.0$. Therefore, we set the balance parameters to $1.0$ in our experiments.

- Compared to changes in $\beta$, variations in $\alpha$ result in more pronounced changes in the model's performance. This indicates that the noisy identification and rectification module contributes more significantly to enhancing the model's performance.

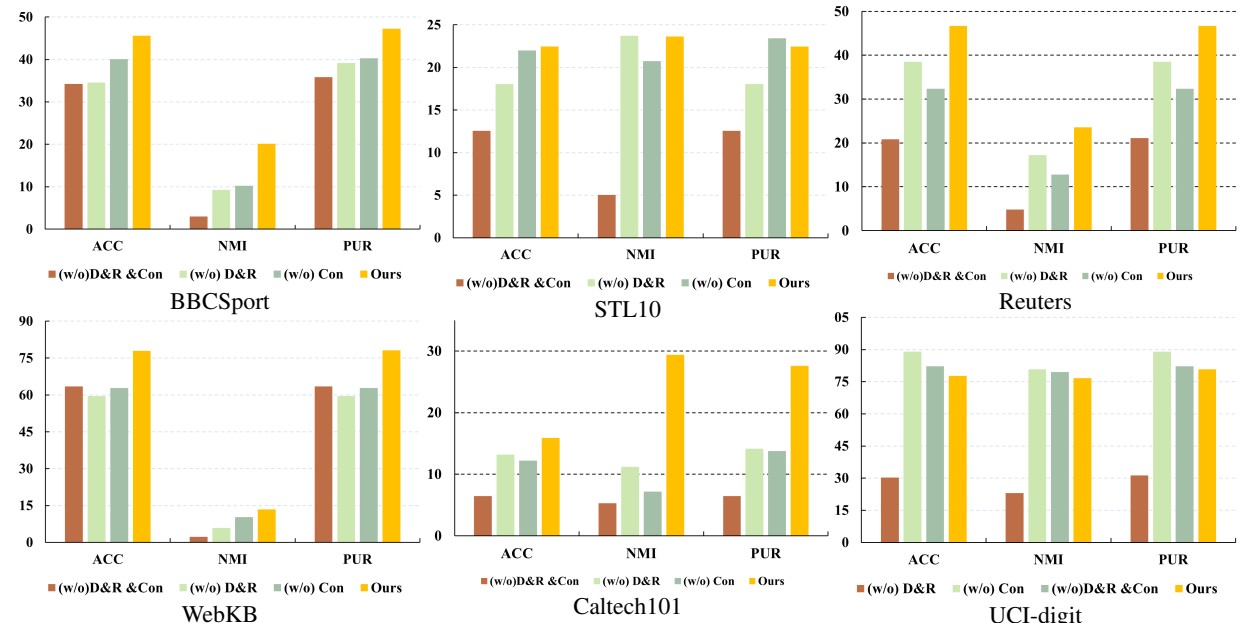

*Figure 9.* Ablation studies on BBCSport, Caltech101, STL10, UCI-digit, WebKB and Reuters datasets with 30% noisy ratio.

### Sensitive analysis of threshold $\tau$:

We conduct experiments to evaluate the influence of the threshold parameter $\tau$. We varied the value of $\tau$ within the range of $\{0.2, 0.4, 0.6, 0.8\}$. The results are demonstrated in Fig. 6. Based on the results, we have the following observation. As the threshold increases, the clustering performance of the model improves progressively. We attribute this to the fact that higher thresholds reduce the number of incorrect contrastive learning sample pairs, thereby enhancing the model's discriminative ability.

### A.3.4. VISUALIZATION ANALYSIS

To gain deeper insights into the representations learned by AIRMVC, we leverage $t$-SNE (Van der Maaten & Hinton, 2008) to visualize the latent feature space. Specifically, we extract the representations using the UCI-digit dataset and map them into a lower-dimensional space for visualization. The results, as shown in Fig. 11, reveal the following observations: as training progresses, the clustering structures in the UCI-digit dataset become progressively more distinct. By the 200th epoch, well-separated and identifiable cluster structures are observed. These results highlight the effectiveness of AIRMVC in uncovering the intrinsic cluster structures of the data.

### A.4. Setting Details of AIRMVC

### Hyper-parameter Settings in our AIRMVC:

In this subsection, we report the statistics summary and hyper-parameter settings of our proposed method in Tab. 7.

### Detailed information about the datasets:

- BBCSport[1]: The BBCSport dataset includes 544 sports news articles categorized into five classes: athletics, cricket, football, rugby, and tennis. Each document is described from three different views with dimensions of 2582, 2544, and 2465. This dataset is essential for research in multi-view learning, text classification, and natural language processing, offering a robust foundation for developing and evaluating advanced algorithms.

- Reuters[2]: Reuters is a collection of documents. It contains 1200 samples, which can be divided into 5 classes. This

---

[1] http://mlg.ucd.ie/datasets/bbc.html
[2] http://archive.ics.uci.edu/dataset/137/reuters+21578+text+categorization+collection

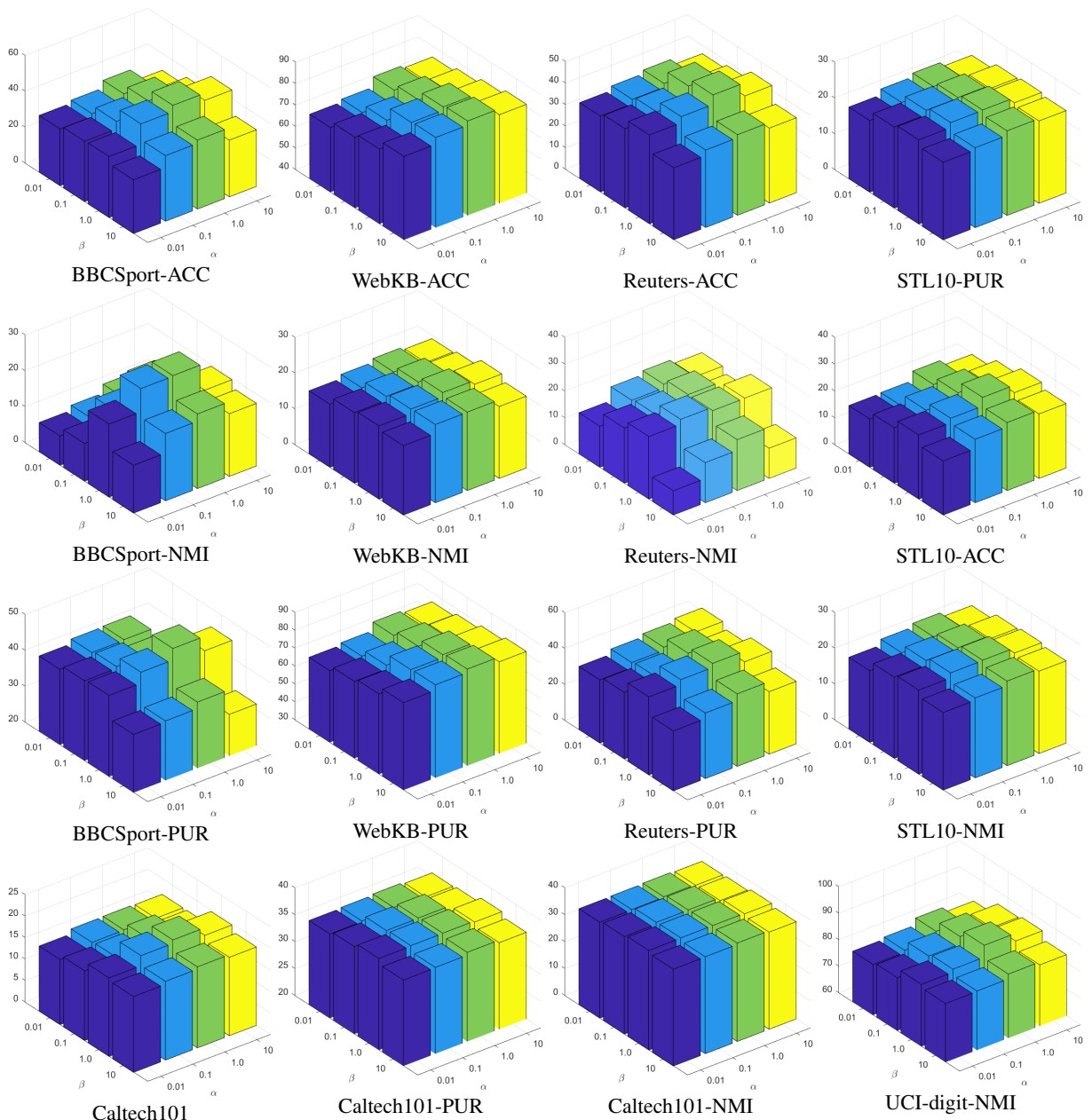

*Figure 10.* Sensitivity Analysis for $\alpha$ and $\beta$ with ACC, NMI, and PUR on BBCSport, WebKB, Reuters, Caltech101, UCI-digit, and STL10 datasets with 10% noisy ratio.

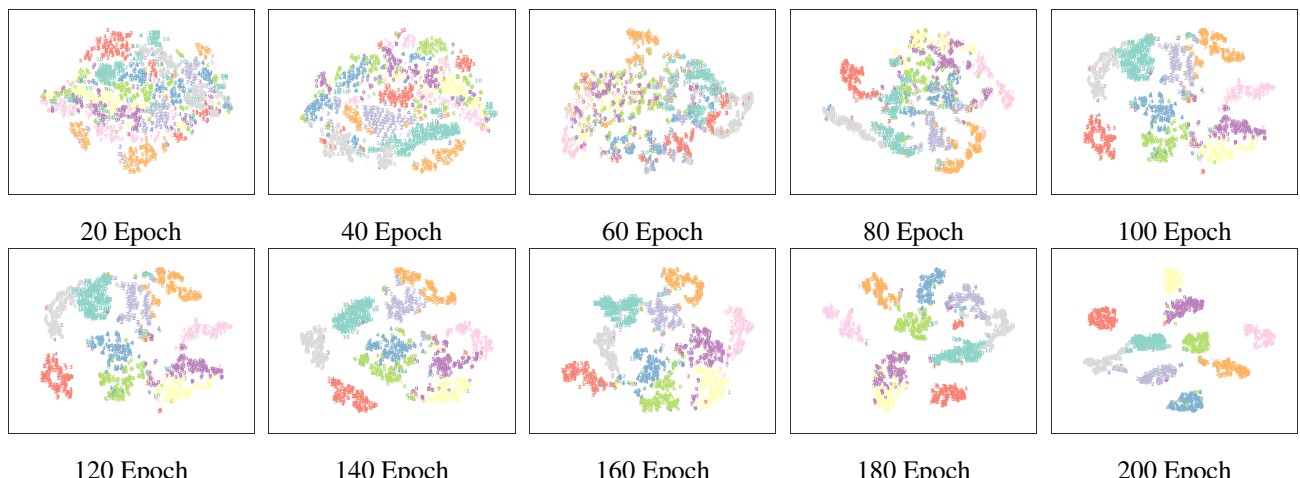

| 20 Epoch | 40 Epoch | 60 Epoch | 80 Epoch | 100 Epoch |
|----------|----------|----------|----------|-----------|
| 120 Epoch | 140 Epoch | 160 Epoch | 180 Epoch | 200 Epoch |

*Figure 11.* Visualization of the representations during the training process on UCI-digit dataset.

*Table 6.* Multi-view clustering performance on six benchmark datasets (Part 3/4). The highest performance metrics are emphasized in red, while the runner-up results are marked in blue.

| | Noisy Rate | | 10% | | | 30% | | | 50% | | | 70% | | |
|---|---|---|---|---|---|---|---|---|---|---|---|---|---|---|
| | Metric | | ACC↑ | NMI↑ | PUR↑ | ACC↑ | NMI↑ | PUR↑ | ACC↑ | NMI↑ | PUR↑ | ACC↑ | NMI↑ | PUR↑ |
| **BBCSport** | SiMVC | CVPR 2021 | 42.83 | 11.90 | 46.88 | 39.52 | 12.48 | 41.54 | 35.29 | 11.02 | 40.07 | 32.35 | 05.09 | 36.76 |
| | CoMVC | CVPR 2021 | 39.71 | 05.38 | 40.62 | 37.50 | 04.03 | 37.13 | 35.48 | 02.13 | 35.85 | 32.17 | 01.62 | 35.48 |
| | MFLVC | CVPR 2022 | 46.87 | 18.69 | 46.87 | 44.12 | 15.49 | 44.12 | 40.51 | 14.44 | 40.51 | 39.15 | 12.18 | 40.43 |
| | DealMVC | ACM MM 2023 | 40.25 | 07.56 | 40.44 | 35.48 | 06.27 | 35.48 | 31.80 | 03.94 | 35.47 | 30.38 | 02.89 | 31.78 |
| | SURE | TPAMI 2023 | 35.26 | 12.80 | 37.24 | 32.35 | 10.27 | 36.51 | 31.55 | 08.23 | 36.47 | 29.27 | 06.84 | 26.31 |
| | AIRMVC | Ours | 49.45 | 28.63 | 49.44 | 45.59 | 20.11 | 47.24 | 45.04 | 23.34 | 53.13 | 40.99 | 16.52 | 45.59 |
| **WebKB** | SiMVC | CVPR 2021 | 77.26 | 13.72 | 69.82 | 62.04 | 10.99 | 65.19 | 60.61 | 08.00 | 63.71 | 55.00 | 06.80 | 56.43 |
| | CoMVC | CVPR 2021 | 77.74 | 10.62 | 69.80 | 66.89 | 08.10 | 64.58 | 56.23 | 07.27 | 58.12 | 54.42 | 05.17 | 55.28 |
| | MFLVC | CVPR 2022 | 77.83 | 19.56 | 78.12 | 64.61 | 09.42 | 64.53 | 62.13 | 07.43 | 62.13 | 58.61 | 03.07 | 58.61 |
| | DealMVC | ACM MM 2023 | 66.03 | 15.02 | 65.27 | 60.32 | 04.85 | 60.32 | 59.66 | 04.63 | 58.47 | 55.57 | 01.92 | 55.57 |
| | SURE | TPAMI 2023 | 67.27 | 11.10 | 67.11 | 62.55 | 7.80 | 63.05 | 55.77 | 06.12 | 58.92 | 53.91 | 05.80 | 57.71 |
| | AIRMVC | Ours | 83.73 | 27.15 | 83.73 | 77.93 | 13.48 | 78.12 | 74.98 | 08.42 | 74.98 | 62.89 | 06.69 | 62.32 |
| **Reuters** | SiMVC | CVPR 2021 | 25.83 | 07.80 | 26.67 | 22.50 | 04.69 | 22.67 | 19.17 | 04.12 | 19.17 | 17.75 | 01.83 | 17.92 |
| | CoMVC | CVPR 2021 | 25.92 | 06.35 | 24.50 | 22.67 | 05.47 | 22.67 | 19.75 | 02.92 | 19.75 | 18.25 | 01.93 | 18.25 |
| | MFLVC | CVPR 2022 | 34.92 | 21.24 | 36.00 | 31.08 | 17.94 | 29.50 | 29.91 | 15.77 | 30.33 | 27.42 | 12.07 | 27.42 |
| | DealMVC | ACM MM 2023 | 30.92 | 14.14 | 30.92 | 28.50 | 10.53 | 28.50 | 27.92 | 06.16 | 27.92 | 26.58 | 05.74 | 26.58 |
| | SURE | TPAMI 2023 | 28.35 | 07.13 | 28.90 | 25.58 | 06.14 | 26.18 | 24.57 | 04.63 | 25.23 | 23.07 | 03.43 | 23.27 |
| | AIRMVC | Ours | 48.25 | 26.34 | 48.25 | 46.67 | 23.54 | 46.67 | 44.83 | 21.72 | 45.58 | 41.75 | 20.91 | 42.33 |

dataset is widely used for text classification, information retrieval, and topic modeling, serving as a benchmark for machine learning. It also aids educational purposes by providing a real-world example for hands-on practice in data analysis and model development, significantly advancing the understanding and solutions for text classification challenges.

- UCI-digit[3]: The UCI-digit Dataset, also known as the Optical Recognition of Handwritten Digits, contains handwritten digit images (0-9) represented as 8x8 grayscale matrices, resulting in 64 features per image. Each image is labeled with the corresponding digit. This dataset is widely used in academic research for evaluating machine learning models in image recognition and classification, making it an invaluable resource for advancing pattern recognition and computer vision methodologies.

- WebKB[4]: The WebKB dataset features web pages from four Wisconsin universities, focusing on both the content and the links between them. This dual-aspect dataset is ideal for research in web mining, hyperlink analysis, and multi-view learning, providing a comprehensive resource for developing and evaluating algorithms that explore relationships and content dynamics in academic web environments.

---

[3]https://cs.nyu.edu/roweis/data.html
[4]https://lig-membres.imag.fr/grimal/data.html

*Table 7.* Hyper-parameters Statistics in AIRMVC.

|  | BBCSport | WebKB | Reuters | UCI-digit | Caltech101 | STL10 |
|---|---|---|---|---|---|---|
| **Views** | 2 | 2 | 5 | 3 | 5 | 4 |
| **Samples** | 544 | 1051 | 1200 | 2000 | 9144 | 13000 |
| **Clusters** | 5 | 2 | 6 | 10 | 102 | 10 |
| **Learning Rate** | 1e-4 | 1e-4 | 1e-5 | 1e-4 | 1e-4 | 1e-4 |
| $\tau$ | 0.8 | 0.8 | 0.8 | 0.8 | 0.8 | 0.8 |
| $\alpha$ | 1.0 | 1.0 | 1.0 | 1.0 | 1.0 | 1.0 |
| $\beta$ | 1.0 | 1.0 | 1.0 | 1.0 | 1.0 | 1.0 |

*Table 8.* Multi-view clustering performance on six benchmark datasets (Part 4/4). The highest performance metrics are emphasized in red, while the runner-up results are marked in blue.

|  |  | Noisy Rate | 10% | | | 30% | | | 50% | | | 70% | | |
|---|---|---|---|---|---|---|---|---|---|---|---|---|---|---|
|  |  | Metric | ACC↑ | NMI↑ | PUR↑ | ACC↑ | NMI↑ | PUR↑ | ACC↑ | NMI↑ | PUR↑ | ACC↑ | NMI↑ | PUR↑ |
| UCI-digit | SiMVC | CVPR 2021 | 30.50 | 30.14 | 31.10 | 26.50 | 19.00 | 26.50 | 23.45 | 18.01 | 23.70 | 19.95 | 11.80 | 20.00 |
| | CoMVC | CVPR 2021 | 44.80 | 42.42 | 45.10 | 34.60 | 33.58 | 34.85 | 32.20 | 31.31 | 32.55 | 28.45 | 27.00 | 29.45 |
| | MFLVC | CVPR 2022 | 89.80 | 82.37 | 89.80 | 60.50 | 63.35 | 60.50 | 47.05 | 48.91 | 47.05 | 31.90 | 28.14 | 31.90 |
| | DealMVC | ACM MM 2023 | 84.85 | 80.98 | 84.85 | 70.80 | 67.38 | 70.80 | 65.55 | 62.12 | 65.55 | 48.90 | 49.82 | 48.90 |
| | SURE | TPAMI 2023 | 34.96 | 14.82 | 34.96 | 31.73 | 13.49 | 33.29 | 28.29 | 11.96 | 29.35 | 26.14 | 10.08 | 27.05 |
| | AIRMVC | Ours | 93.95 | 89.08 | 93.95 | 77.60 | 76.65 | 80.75 | 76.05 | 68.17 | 76.05 | 69.20 | 60.38 | 69.20 |
| Caltech101 | SiMVC | CVPR 2021 | 14.88 | 10.98 | 16.28 | 11.72 | 6.55 | 12.98 | 10.79 | 5.57 | 11.94 | 9.63 | 5.05 | 10.97 |
| | CoMVC | CVPR 2021 | 15.00 | 12.92 | 15.80 | 14.34 | 12.26 | 15.49 | 10.04 | 11.66 | 10.42 | 9.58 | 6.53 | 9.95 |
| | MFLVC | CVPR 2022 | 19.63 | 20.27 | 21.78 | 14.87 | 16.52 | 14.33 | 10.59 | 13.76 | 10.79 | 9.60 | 6.80 | 9.63 |
| | DealMVC | ACM MM 2023 | 19.90 | 20.73 | 22.07 | 13.09 | 11.96 | 17.80 | 10.14 | 5.55 | 10.14 | 10.05 | 5.03 | 10.08 |
| | SURE | TPAMI 2023 | 7.44 | 17.91 | 16.29 | 7.09 | 15.85 | 16.03 | 6.84 | 14.57 | 15.01 | 6.33 | 12.58 | 10.62 |
| | AIRMVC | Ours | 21.45 | 37.16 | 34.69 | 15.89 | 29.39 | 27.60 | 11.80 | 23.66 | 23.33 | 11.89 | 21.14 | 20.14 |
| STL10 | SiMVC | CVPR 2021 | 19.51 | 11.33 | 19.52 | 17.52 | 6.83 | 18.06 | 15.91 | 5.45 | 16.04 | 13.44 | 3.62 | 14.00 |
| | CoMVC | CVPR 2021 | 27.07 | 21.49 | 27.36 | 20.78 | 15.57 | 20.70 | 17.68 | 16.15 | 17.75 | 15.37 | 13.92 | 15.56 |
| | MFLVC | CVPR 2022 | 24.39 | 23.34 | 24.39 | 21.14 | 20.19 | 21.47 | 20.58 | 21.92 | 20.14 | 19.08 | 18.24 | 19.08 |
| | DealMVC | ACM MM 2023 | 24.59 | 23.51 | 24.59 | 21.23 | 22.83 | 21.33 | 20.56 | 21.78 | 20.56 | 19.12 | 18.27 | 19.12 |
| | SURE | TPAMI 2023 | 21.14 | 6.72 | 21.66 | 19.36 | 5.87 | 20.76 | 18.34 | 5.64 | 19.11 | 17.33 | 5.45 | 18.15 |
| | AIRMVC | Ours | 28.81 | 25.04 | 29.01 | 22.46 | 23.64 | 22.46 | 21.95 | 23.09 | 22.02 | 20.14 | 22.66 | 20.14 |

- SUNRGBD[5]: The SUNRGBD dataset, essential for indoor scene understanding, contains 10,335 samples across 45 categories, including RGB images, depth images, and 3D point cloud data. Each sample has detailed semantic annotations, covering object categories and spatial locations. This multi-modal dataset supports tasks like scene understanding, object detection, semantic segmentation, and 3D reconstruction. Researchers use SUNRGBD to develop and evaluate algorithms that integrate multi-modal information, advancing indoor scene analysis and computer vision research.

- STL10[6]: The STL10 dataset is a key benchmark for unsupervised feature learning, deep learning, and self-taught learning. It contains 13,000 labeled images across 10 classes and 100,000 unlabeled images, each at 96x96 pixels with four views per image. This dataset enables model pre-training on unlabeled data, simulating real-world scenarios, and is crucial for evaluating feature learning and generalization capabilities, advancing representation learning and machine learning research.

- Caltech101[7]: The Caltech101 dataset is a widely used benchmark in machine learning and computer vision, consisting of 9,144 samples categorized into 102 distinct clusters. It is characterized by its diversity and complexity, featuring five distinct views that provide complementary information for each sample. These views typically include different feature types or representations, making it a suitable dataset for evaluating multi-view clustering and representation learning methods. Due to its rich and heterogeneous structure, Caltech101 has become a standard testbed for assessing the performance of clustering algorithms in handling multi-view and high-dimensional data.

**Algorithm:** Due to the limited space of the original paper, we present the algorithm in this part. The detailed training

---

[5] https://rgbd.cs.princeton.edu/
[6] https://cs.stanford.edu/~acoates/stl10/
[7] https://data.caltech.edu/records/mzrjq-6wc02

---

**Algorithm 1** Training Algorithm of our designed AIRMVC

---

**Input**: The multi-view dataset $\{x^v\}_{v=1}^V$; the epochs $e$; batch size $B$

**Output**: The clustering result **R**.

1: **for** 1 to $e$ **do**
2:     **E-Step:** update the parameters $\{(\mu_k, \sigma_k)\}_{k=1}^K$ in AIRMVC and the $\{\varphi_i\}_{i=1}^N$
3:     **M-Step:**
4:     **repeat**
5:       Obtain the representations **E** by the encoder network.
6:       Identify the noisy data with Eq. (6).
7:       Rectify the noisy data with Eq. (7).
8:       Calculate the rectification loss, contrastive loss, reconstruction loss with Eq. (8), (11), (12).
9:       Calculate the total loss $\mathcal{L}$ by Eq.(13).
10:       Update model by minimizing $\mathcal{L}$ with Adam optimizer.
11:     **until** the epoch finishes
12: **end for**
13: **Output** The clustering results **R**.

---

*Table 9.* Multi-view clustering performance on six benchmark datasets with 90% noise ratio. The highest performance metrics are emphasized in red, while the runner-up results are marked in blue.

| Dataset | | BBCSport | | | WebKB | | | Reuters | | |
|---|---|---|---|---|---|---|---|---|---|---|
| **Metrics** | | **ACC↑** | **NMI↑** | **PUR↑** | **ACC↑** | **NMI↑** | **PUR↑** | **ACC↑** | **NMI↑** | **PUR↑** |
| **SiMVC** | **CVPR 2021** | 29.78 | 04.15 | 35.85 | 52.43 | 04.60 | 56.47 | 16.92 | 01.08 | 17.00 |
| **CoMVC** | **CVPR 2021** | 30.33 | 01.59 | 32.75 | 51.67 | 03.85 | 50.15 | 16.67 | 01.41 | 16.67 |
| **MFLVC** | **CVPR 2022** | 37.13 | 07.14 | 32.52 | 56.14 | 01.68 | 56.14 | 21.33 | 04.35 | 21.33 |
| **DealMVC** | **ACM MM 2023** | 29.78 | 02.76 | 30.47 | 50.77 | 01.28 | 50.77 | 20.87 | 03.58 | 20.87 |
| **SURE** | **TPAMI 2023** | 28.20 | 03.13 | 24.26 | 52.98 | 04.20 | 56.61 | 21.18 | 02.82 | 21.75 |
| **CANDY** | **NeurIPS 2024** | 15.26 | 01.42 | 33.48 | 14.27 | 02.39 | 14.27 | 17.84 | 04.70 | 26.50 |
| **RMCNC** | **TKDE 2024** | 23.90 | 03.78 | 23.87 | 51.87 | 03.47 | 50.48 | 20.17 | 05.23 | 20.25 |
| **TGM-MVC** | **ACM MM 2024** | 17.38 | 03.57 | 17.38 | 50.24 | 05.21 | 50.24 | 20.58 | 03.87 | 20.58 |
| **SCE-MVC** | **NeurIPS 2024** | 37.62 | 08.47 | 37.28 | 51.67 | 01.20 | 51.67 | 16.67 | 04.76 | 16.67 |
| **MVCAN** | **CVPR 2024** | 23.34 | 01.79 | 24.68 | 57.71 | 03.05 | 57.15 | 19.41 | 04.97 | 19.83 |
| **DIVIDE** | **AAAI 2024** | 26.29 | 02.04 | 37.32 | 50.05 | 03.70 | 54.34 | 24.08 | 06.25 | 26.42 |
| **AIRMVC** | **Ours** | **39.52** | **09.33** | **40.26** | **59.18** | **05.36** | **60.08** | **33.50** | **20.60** | **35.35** |
| Dataset | | UCI-digit | | | Caltech101 | | | STL10 | | |
| **Metrics** | | **ACC↑** | **NMI↑** | **PUR↑** | **ACC↑** | **NMI↑** | **PUR↑** | **ACC↑** | **NMI↑** | **PUR↑** |
| **SiMVC** | **CVPR 2021** | 15.55 | 08.64 | 16.00 | 07.78 | 03.16 | 09.61 | 11.72 | 01.71 | 12.66 |
| **CoMVC** | **CVPR 2021** | 24.15 | 26.41 | 26.95 | 08.61 | 05.55 | 10.56 | 12.63 | 08.21 | 12.75 |
| **MFLVC** | **CVPR 2022** | 29.25 | 29.18 | 29.25 | 08.43 | 05.29 | 08.98 | 18.04 | 13.93 | 18.04 |
| **DealMVC** | **ACM MM 2023** | 37.40 | 39.04 | 37.45 | 09.96 | 04.03 | 09.38 | 18.45 | 17.32 | 18.45 |
| **SURE** | **TPAMI 2023** | 23.92 | 08.83 | 25.13 | 06.21 | 10.41 | 06.85 | 16.93 | 04.97 | 17.59 |
| **CANDY** | **NeurIPS 2024** | 51.05 | 42.16 | 53.70 | 10.00 | 10.60 | 16.56 | 16.32 | 15.44 | 16.72 |
| **RMCNC** | **TKDE 2024** | 19.15 | 08.81 | 21.15 | 06.56 | 10.89 | 06.10 | 13.35 | 01.26 | 13.57 |
| **TGM-MVC** | **ACM MM 2024** | 49.83 | 63.42 | 55.59 | 05.45 | 17.53 | 16.21 | 17.74 | 15.66 | 17.24 |
| **SCE-MVC** | **NeurIPS 2024** | 57.70 | 49.43 | 57.15 | 10.08 | 18.16 | 18.41 | 13.85 | 10.93 | 13.59 |
| **MVCAN** | **CVPR 2024** | 57.25 | 50.87 | 57.48 | 9.07 | 13.49 | 3.77 | 16.15 | 15.35 | 16.33 |
| **DIVIDE** | **AAAI 2024** | 57.05 | 50.09 | 57.75 | 9.02 | 15.85 | 13.28 | 18.09 | 12.00 | 18.63 |
| **AIRMVC** | **Ours** | **58.20** | **51.94** | **58.20** | **10.85** | **19.01** | **18.96** | **19.90** | **19.92** | **19.90** |

process is presented in Algorithm. 1

## A.5. Comparative Algorithms

In this paper, we compare our AIRMVC with 11 baselines, including CoMVC (Trosten et al., 2021), SiMVC (Trosten et al., 2021), MFLVC (Xu et al., 2022b), DealMVC (Yang et al., 2023b), SURE (Yang et al., 2023a), CANDY (Guo et al.), TGM-MVC (Wang et al., 2024a), SCE-MVC (Wang et al., 2024b) RMCNC (Sun et al., 2024), and MVCAN (Xu et al., 2024). We list the specific information of those methods as follows:

- CoMVC & SiMVC (Trosten et al., 2021): CoMVC and SiMVC are deep multi-view clustering methods that avoid aligning all representations simultaneously. These methods incorporate a contrastive learning strategy to enhance clustering performance.

- MFLVC (Xu et al., 2022b): MFLVC introduces two consistency objectives for deep multi-view clustering by employing contrastive learning strategies at both the high-level feature space and the semantic label space.

- DealMVC (Yang et al., 2023b): DealMVC proposes a dual-calibration mechanism tailored for deep multi-view clustering. Its calibration contrastive loss effectively distinguishes between similar yet distinct samples across multiple views.

- SURE (Yang et al., 2023a): SURE addresses challenges related to incomplete multi-view data by developing a noise-robust contrastive loss. This approach is theoretically analyzed to handle partially unaligned views and missing samples.

- CANDY (Guo et al.): CANDY leverages inter-view similarities as context to uncover false negatives and integrates a spectral-based module for denoising correspondence in multi-view scenarios.

- TGM-MVC (Wang et al., 2024a): TGM-MVC employs a tree-based framework to effectively capture the heterogeneity between different views, offering a unique perspective on multi-view clustering.

- SCE-MVC (Wang et al., 2024b): SCE-MVC adopts game-theoretic principles and Shapley values to address multi-view clustering challenges, providing an innovative solution to enhance clustering robustness.

- RMCNC (Sun et al., 2024): RMCNC introduces a noise-tolerant contrastive loss, designed to mitigate the impact of misaligned pairs in multi-view clustering tasks.

- MVCAN (Xu et al., 2024): MVCAN employs an unshared network structure and implements a two-level optimization strategy, enabling improved clustering performance in multi-view scenarios.

- DIVIDE (Lu et al., 2024): DIVIDE utilizes a random walk approach to progressively identify reliable data pairs, enhancing the stability and robustness of contrastive learning in multi-view clustering.

