# OpenReview forum: "Automatically Identify and Rectify: Robust Deep Contrastive Multi-view Clustering in Noisy Scenarios"
_ICML.cc/2025/Conference — ICML 2025 spotlightposter_

### Official Review · Reviewer_yWYk · 2025-03-04

**Overall Recommendation:** 4

**Summary:**

In this paper, the authors propose a novel multi-view clustering method AIRMVC for noisy scenarios. They formulate the noisy identification as the anomaly problem. Besides, a noise-robust contrastive loss is designed to enhance the model performance. Experiments on six datasets show the effectiveness of the proposed method.

**Claims And Evidence:**

The motivation of AIRMVC is clearly articulated and supported by experimental validation. Additionally, extensive supplementary experiments are provided in the appendix to further reinforce the findings.

**Essential References Not Discussed:**

The related work is well-presented and the literature review is thorough.

**Experimental Designs Or Analyses:**

The authors conducted extensive experiments, with most baseline methods being from 2024, effectively demonstrating the efficacy of the proposed approach.

**Methods And Evaluation Criteria:**

The authors investigate the problem of multi-view clustering in noisy scenarios, a challenge widely encountered in real-world applications. The proposed method is well-aligned with the stated motivation and is designed to enhance model robustness under noisy conditions, offering practical insights for real-world implementations.

**Other Comments Or Suggestions:**

1.The notation II in Equation (10) should be explicitly defined for clarity.

2.On page 7, line 364, there is a typographical error in the quotation marks for "w/o D&R&Con."

**Other Strengths And Weaknesses:**

Strengths:

1.The transformation of noise identification into an anomaly detection problem is an intriguing approach.

2.This paper conducts extensive experiments comparing the proposed method with 2024 state-of-the-art approaches, and the comprehensive experimental results validate its effectiveness.

3.Theoretical analysis demonstrates the robustness of the proposed contrastive learning mechanism in noisy environments.

Weaknesses:

1.In the noise identification module, both the projector and classifier are designed. Are these components shared across multi-views, or do they operate independently? The authors should provide a clear explanation regarding this aspect.

2.The authors have not discussed the limitations of AIRMVC or outlined potential directions for future research.

3.Although the authors conducted explanatory experiments in Figure 3 to support their motivation, they only utilized the relatively small-scale BBCSport dataset. It is recommended to perform experiments on larger datasets to further validate the motivation’s effectiveness.

4.The authors should release the source code to enhance the reproducibility of the study.

**Questions For Authors:**

Please see Weaknesses and suggestions.

**Relation To Broader Scientific Literature:**

The paper presents a comprehensive and well-rounded literature review.

**Theoretical Claims:**

The authors provide a theoretical proof for noise-robust contrastive learning for supporting the rationale behind this approach.

---

> ### Author Rebuttal · Authors · 2025-03-31
>
> **Explanation for projector and classifier:** Thanks. We perform feature mapping and transformation in the latent space using a projector, and the sample predictions are obtained through a classifier. The projector and classifier are shared across different views. We will include the corresponding description in the final version.
>
> **Limitations & Future directions of AIRMVC:** Thanks. In AIRMVC, we made an initial attempt to identify and rectify noise in an unsupervised setting and designed a robust contrastive learning method to further enhance the robustness of the model. However, the correction process relies heavily on the accuracy of the predicted distribution, which is the primary limitation of AIRMVC. In the future, improving the accuracy of the predicted distribution and exploring other reliable supervisory signals will be promising research directions.
>
> **Large-scale motivation experiments:** Thanks. Following your suggestions, we conducted experiments on the Caltech101 and STL10 datasets with a 10% noise ratio. The experimental results are presented in Tab.1 and Tab.2. From these results, we observe the same conclusions as those described in the submitted version, presented as follows:
>
> 1) Simply merging noisy multi-view data results in the most degraded clustering performance. This is primarily due to the absence of a noise rectification mechanism, which causes the negative effects of noisy views to compound each other. Moreover, the fusion process intensifies the influence of noise, leading to a scenario where multi-view clustering performs even worse than using a single view alone.
>
> 2) In comparison to directly correcting noise based on the first view, our proposed AIRMVC demonstrates superior performance. This advantage arises because direct correction from a single view tends to enforce uniformity across views, potentially suppressing essential complementary information. In contrast, our noise detection and rectification strategy effectively removes noisy samples from each view while preserving beneficial cross-view diversity, thereby enhancing the overall clustering performance.
>
> Tab.1 Motivation experiments on STL10 dataset.
>
> | Metric |  Ours  | Directly Rectify | Single View | Noisy data |
> |:------:|:------:|:----------------:|:-----------:|:----------:|
> |   ACC  | 28.81  |      26.41       |    22.22    |   15.05    |
> |   NMI  | 25.04  |      24.25       |    19.04    |   10.78    |
> |   PUR  | 29.01  |      26.80       |    23.18    |   13.78    |
>
> Tab.2 Motivation experiments on Caltech101 dataset.
>
> | Metric |  Ours  | Directly Rectify | Single View | Noisy data |
> |:------:|:------:|:----------------:|:-----------:|:----------:|
> |   ACC  | 21.45  |      18.62       |    15.26    |   11.25    |
> |   NMI  | 37.16  |      30.82       |    22.29    |   18.61    |
> |   PUR  | 34.69  |      29.52       |    20.18    |   16.28    |
>
> **Code:** Thanks. Following your suggestion, we will release the code in the final version.
>
> **Notation & Typos:** Thanks. We will add the notations of Eq.10 and correct the typos in page 7. Furthermore, we will review the entire paper to enhance the overall presentation.

---

> > ### Comment · Reviewer_yWYk · 2025-04-02
> >
> > Thank for rebuttal from the authors. My concerns and confusions are well-addressed and thus I would like to increase my score.

---

> > > ### Author Response · Authors · 2025-04-05
> > >
> > > Dear Reviewer,
> > >
> > > Thanks for your increasing the score. We greatly appreciate the time and effort you have dedicated to reviewing our work!
> > >
> > > Sincerely,
> > >
> > > The Authors

---

### Official Review · Reviewer_jA5D · 2025-03-05

**Overall Recommendation:** 3

**Summary:**

This paper addresses the challenge of noisy data in multi-view clustering by proposing a method called AIRMVC. Specifically, AIRMVC first formulates noise identification and employs a Gaussian Mixture Model (GMM) to achieve this. It then introduces a hybrid rectification strategy with an interpolation mechanism to mitigate the adverse effects of noisy data. The paper validates the effectiveness of AIRMVC on six multi-view clustering datasets.

**Claims And Evidence:**

Not all claims are fully supported. For instance, the paper asserts that no prior work has developed dedicated frameworks for identifying and rectifying noisy data. However, MVCAN (Xu et al., 2024) appears to be an earlier attempt at addressing this issue but is not appropriately acknowledged.

**Essential References Not Discussed:**

Although related works are cited, the paper lacks a sufficient discussion on MVCAN (Xu et al., 2024), which may lead to an overstatement of its contributions.

**Experimental Designs Or Analyses:**

Not all experiments are well-designed. Firstly, the datasets used are relatively small (fewer than 13,000 samples), which limits the evaluation of the method’s scalability and generalizability. Secondly, the evaluations are conducted with hand-crafted noise rather than real-world noise, potentially affecting the practical applicability of the method.

**Methods And Evaluation Criteria:**

While the method is designed to address the noisy view problem, the chosen datasets and experimental settings are not sufficiently justified.

**Other Comments Or Suggestions:**

I look forward to seeing the method evaluated on large-scale datasets with real-world noise for a more comprehensive assessment of its effectiveness.

**Other Strengths And Weaknesses:**

Strengths:

The proposed method achieves state-of-the-art performance on the six selected datasets.

Weaknesses:

The novelty of the paper is questionable, as it overclaims its contribution to handling noisy views. MVCAN (Xu et al., 2024) may already have laid the groundwork for this problem.

The experimental evaluation is insufficient, relying on small-scale datasets and artificially introduced noise instead of real-world noisy data.

I would consider improving my rating if the authors could give more clarifications or experiment results to address my concerns.

**Questions For Authors:**

Please see the weaknesses

**Relation To Broader Scientific Literature:**

The paper designs a method for handling the noisy view issue in the multi-view clustering task.

**Theoretical Claims:**

No

---

> ### Author Rebuttal · Authors · 2025-03-31
>
> **Additional experiments:** Thanks. Following your suggestions, with NVIDIA A6000 GPU we conduct experiments on CIFAR10 dataset, which contains 60,000 samples, 4 views and 10 classes. Besides, YouTube is a comprehensive video platform. We extract facial images from videos as a real-world data source. These multi-view facial images may include low-quality samples, which are treated as noise in our analysis. The Youtube dataset comprises 38,654 samples, 4 views and 10 classes. Detailed statistic information of the datasets is demonstrated in Tab.1. From the results shown in Tab.2 and Tab.3, we conclude that AIRMVC could achieve reliable performance on both large-scale dataset and real-world dataset, demonstrating its generalization capability.
>
> Tab.1 Statistic information of the datasets.
> | Datasets | Class | Sample | View |
> |:--------:|:-----:|:------:|:----:|
> |  Youtube |   10  | 38,654 |   4  |
> |  CIFAR10 |   10  | 60,000 |   4  |
>
> Tab.2 Experiment on YouTube dataset. OOM  denotes out-of-memory during training process.
> | Metric |  CANDY |  RMCNC | TGM-MVC | SCE-MVC | MVCAN | DIVIDE |  Ours  |
> |:------:|:------:|:------:|:-------:|:-------:|:-----:|:------:|:------:|
> |   ACC  | 62.86  | 53.05  |  58.26  |  60.54  |  OOM  | 60.16  | 66.23  |
> |   NMI  | 70.06  | 65.27  |  55.91  |  64.22  |  OOM  | 65.38  | 70.94  |
> |   PUR  | 70.20  | 63.81  |  60.12  |  65.54  |  OOM  | 63.01  | 75.10  |
>
> Tab.3 Experiment on CIFAR dataset.
> | Noisy   Rate |   0.1  |   0.1  |   0.1  |   0.3  |   0.3  |   0.3  |   0.5  |   0.5  |   0.5  |   0.7  |   0.7  |   0.7  |
> |:------------:|:------:|:------:|:------:|:------:|:------:|:------:|:------:|:------:|:------:|:------:|:------:|:------:|
> |    Metric    |   ACC  |   NMI  |   PUR  |   ACC  |   NMI  |   PUR  |   ACC  |   NMI  |   PUR  |   ACC  |   NMI  |   PUR  |
> |     CANDY    | 20.16  | 12.35  | 21.21  | 18.25  | 11.82  | 18.01  | 16.04  |  9.54  | 16.05  | 14.17  |  9.06  | 14.64  |
> |     RMCNC    | 19.25  | 11.52  | 20.58  | 18.26  | 10.64  | 19.25  | 16.45  |  8.68  | 16.25  | 15.05  |  8.14  | 14.99  |
> |    TGM-MVC   | 17.82  | 10.02  | 18.99  | 15.29  |  7.91  | 14.25  | 13.42  |  6.04  | 13.57  | 11.05  |  5.71  | 12.52  |
> |    SCE-MVC   | 18.25  | 10.55  | 19.54  | 18.02  | 10.00  | 18.57  | 15.15  |  8.02  | 16.05  | 14.23  |  7.64  | 13.16  |
> |     MVCAN    |   OOM  |   OOM  |   OOM  |   OOM  |   OOM  |   OOM  |   OOM  |   OOM  |   OOM  |   OOM  |   OOM  |   OOM  |
> |    DIVIDE    | 20.57  | 11.26  | 21.27  | 18.69  | 10.06  | 19.95  | 16.84  |  8.32  | 17.62  | 14.05  |  6.05  | 14.85  |
> |     Ours     | 22.62  | 13.71  | 23.34  | 21.67  | 13.24  | 22.52  | 20.08  | 12.49  | 20.83  | 17.68  | 10.25  | 18.26  |
>
> **Discussion with MVCAN:** Thanks. We discuss AIRMVC with MVCAN from three key perspectives:
>
> 1) Optimization Strategy: MVCAN adopts a two-level iterative optimization framework, consisting of T-level and R-level optimization to refine the network. In contrast, In contrast, AIRMVC focuses on noise detection and correction using a Gaussian Mixture Model (GMM) and directly optimizes the network.
>
> 2) Soft Label Acquisition: MVCAN employs a parameter-decoupled model to obtain view-specific representations and soft labels, mitigating the influence of noisy views. AIRMVC leverages a GMM trained with a shared projector and classifier to generate soft labels.
>
> 3) Module Design: MVCAN incorporates unshared parameters, distinct clustering optimization functions, and a two-level iterative optimization approach. In comparison, AIRMVC introduces a dedicated noise detection and correction mechanism, along with a noise-robust contrastive learning framework to enhance model robustness.
>
> **Novelty of AIRMVC:** Thanks. The novelty of AIRMVC mainly contains the following perspectives.
>
> 1) Leveraging GMM, we reformulate the noise identification as an anomaly identification problem and propose a hybrid rectification strategy to automatically correct the noisy data.
>
> 2) We design a noise-robust contrastive mechanism to generate more reliable representations. Theoretically, we have demonstrated that the features generated by this mechanism are more beneficial for downstream tasks.
>
> 3) Extensive experiments on different benchmark datasets to verify the effectiveness and robustness of AIRMVC.

---

> > ### Comment · Reviewer_jA5D · 2025-04-03
> >
> > Thank you for the detailed responses. I appreciate the additional experiments on large-scale and real-world noisy datasets, which address my second major concern. Furthermore, the detailed discussion on MVCAN highlights the novelty of the proposed method. I would like to raise my rating and maintain a positive stance on the paper.

---

> > > ### Author Response · Authors · 2025-04-07
> > >
> > > Dear Reviewer,
> > >
> > > Thank you for increasing the score and acknowledging our approach. We sincerely appreciate the time and effort you dedicated to reviewing our work. Based on your suggestions, we will make further improvements in the final version.
> > >
> > > Best,
> > >
> > > The Authors

---

### Official Review · Reviewer_6d3k · 2025-03-07

**Overall Recommendation:** 3

**Summary:**

The paper considers the problem of multi-view clustering in the presence of noise. In particular, a new approach is proposed that aims to detect noisy samples, characterized as outliers, and to rectify them based on the assumption that the first view is noise-free. In addition, the construction of the pairs in the contrastive loss is improved by taking into consideration the soft clustering labels.

## Update after rebuttal
After the additional clarifications provided by the authors, I have decided to increase my rating. While it still is based on relatively strong assumptions, which should be thoroughly discussed in a limitation section if the paper would be accepted, it could serve as an initial exploration of this setting.

**Claims And Evidence:**

Yes

**Essential References Not Discussed:**

Overall, the reviewer believes that prior work is cited adequately, but believes that the paper would benefit from a clearer discussion of what noise problems the different baselines address, as the baselines are designed for different types of noise.

**Experimental Designs Or Analyses:**

As mentioned above, the experimental design is somewhat unclear when it comes to the addition of noise and clarifications are required. Beyond this, the experimental design appears sound. However, it would be beneficial to also report the clean performance for reference, despite the focus being mostly on the noisy setting.

**Methods And Evaluation Criteria:**

While the setup is mostly reasonable, the type of noise added to the samples is not described. Based on the introduction and problem definition, the noise considered in this work is due to the image being corrupted, while previous referenced work mostly considers noise in the sense of alignment (are the two views representing the same object). However, how this noise has been added and how the X% of data in the experiments has been corrupted is unclear.

Related to this point, the proposed approach builds on the strong assumption of having one clean view for the rectification step. While the authors state that this is done following previous work with references provided, these prior works, to the reviewers knowledge make a different assumption where some data is known to be aligned and some misaligned and do not assume that there is one completely uncorrupted view. What is the effect if this assumption is not valid?

In summary, additional clarifications on this setup are needed to ensure that the evaluation is fair.

**Other Comments Or Suggestions:**

The presentation of the problem formulation seems to have been moved out of Sec. 3, while it still is mentioned in Line 139 (can be removed).
In line 304, "sub-optimal" should maybe be "runner-up" or "second best".
For Table 4, state explicitly that this is the 10% noise scenario.

**Other Strengths And Weaknesses:**

Overall, the paper addresses the interesting problem of corrupted views in deep multi-view clustering and the paper is mostly well-written and presents a set of relevant ablation studies to highlight the necessity of the different components. However, the presentation of the problem as well as how this relates to previous works on robustness in the multi-view space, which generally focus on another type of noise, should be improved. In addition it is based on a key assumption (first view is noise-free) and the effect of this assumption should be discussed, as it appears to differ from the assumptions in prior works and thus benefits the proposed approach.

**Questions For Authors:**

Please elaborate the experimental setup and comment on the effect of the assumption on the first view.
The work further leverages another assumption, which is the presence of balanced clusters in 3.1. Does this overly bias the model to clustering settings where there are balanced classes?

**Relation To Broader Scientific Literature:**

Within the extended multi-view clustering literature, there has been some work on designing more robust approaches. However, these have mostly been focusing on the design of approaches that are able to handle partial view alignment or incomplete views. While there are certain approaches that aim to address robustness to noise (such as Xu et al, CVPR 2024), it is a less explored domain and the paper contributes a new approach toward it.

**Theoretical Claims:**

The paper includes a theoretical interpretation of the noise-robust loss in Theorem 4.1, which appears to be correct.

---

> ### Author Rebuttal · Authors · 2025-03-31
>
> **Explanation for adding noise:** Thanks. Different with the noisy alignment, we simulate noisy scenarios by injecting standard Gaussian noise to the original views, excluding the first view. Specifically, we generate random Gaussian noise with the same shape as the view and inject it into the original views at a ratio of x%. The parameter x% scales the generated Gaussian noise, thereby simulating different levels of noise contamination.
>
> **Experiments on clean data:** Thanks. Following your suggestion, we conduct experiments on clean data with six datasets. The results are shown in Tab.1. Due to character limitations, more results can be found in Tab.1 at https://anonymous.4open.science/r/Res-11B2. From the results, we find that AIRMVC achieves promising performance in both clean and noisy scenarios.
>
> Tab.1 Clean data performance
> |Datasets|UCI-digit|-|-|WebKB|-|-|STL10|-|-|
> |-|-|-|-|-|-|-|-|-|-|
> |Metric|ACC|NMI|PUR|ACC|NMI|PUR|ACC|NMI|PUR|
> |CANDY|85.45|77.99|85.45|35.15|10.55|34.74|28.15|22.68|28.15|
> |RMCNC|40.51|23.16|35.68|79.05|21.84|79.99|23.05|15.28|24.64|
> |TGM-MVC|64.35|65.76|69.40|79.64|16.61|79.64|28.18|20.86|28.51|
> |SCE-MVC|84.55|76.48|86.15|78.65|19.04|77.54|28.64|24.59|29.05|
> |DIVIDE|89.25|81.52|89.45|69.61|20.20|78.12|29.68|23.63|28.95|
> |Ours|94.55|90.10|94.55|80.65|21.54|80.65|30.26|24.88|30.95|
>
> **Explanation for assumption:** Thanks. We provide explanation from three perspectives.
>
>  1) Different with noisy data align, AIRMVC extends the definition of noisy by considering the presence of noise within views. Since we propose a method for detecting and correcting noise, an "ideal" view is required as a reference standard. In the unsupervised multi-view clustering scenario, there is no available label information. Therefore, we assume that the first view serves as an ideal view, acting as pseudo-supervision to correct noise in the other views.
>
> 2) We show this assumption with a real-world multi-view scenario, i.e., the ideal view supplements and corrects the other views. For example, consider a case where the first view consists of high-resolution images, while the second view consists of low-resolution images. In the field of super resolution field, it is common to use high-resolution images (ideal view) to supplement information and guide the learning of low-resolution images (other view), e.g., 2019-ICCV-Guided Super-Resolution as Pixel-to-Pixel Transformation and 2021-CVPR-Robust Reference-based Super-Resolution via C2-Matching. Similarly, we select one view as the reference ("ideal") view to supplement and correct the other views.
>
> 3) Previous works have regarded data partially align as noisy. During the model's testing phase, they use an alignment strategy to align the $v-1$ views to the first view, thereby fusing multi-view feature for clustering, e.g., CANDY (line 53 of https://github.com/XLearning-SCU/2024-NeurIPS-CANDY/blob/main/model.py) and RMCNC (line 236 of https://github.com/sunyuan-cs/2024-TKDE-RMCNC/blob/main/RMCNC_main/sure_inference.py). This alignment operation implies that these papers consider the first view as an ideal view. Therefore, although the scenario settings may differ, to maintain generality, we also treat the first view as an ideal view.
>
> **Explanation for baselines:** Thanks. Previous studies consider data partially align as noisy. In our work, we extend the definition of noise and explore a more common noisy scenario, where noise exists within individual views. Recently, MVCAN is the only work that explores the issue of noisy views, leaving no other methods available for direct comparison. MVCAN incorporated comparisons with numerous contrastive learning-based methods. Following this setup of MVCAN, we evaluated the performance of various algorithms under our proposed noisy setting. To further validate the effectiveness of our approach, we included the latest multi-view clustering methods from 2024 in Tables 1 and 2 of our submitted version. Moreover, our selection of a substantial number of contrastive learning-based methods is that contrastive learning could enhance both the model's robustness and discriminative capability. Therefore, in the absence of directly comparable methods, we select contrastive learning-based methods to demonstrate the effectiveness of our method.
>
> **Explanation for balanced clusters:** Thanks. Cluster balance is a widely adopted default assumption in clustering problems, and we follow this common assumption as well. Additionally, to further verify the cluster balance of samples, we conduct statistical analyses on the datasets used in AIRMVC. The results indicate that the sample classes in the utilized datasets are nearly balanced. Due to space limitations, detailed results can be found in Tab.2 in https://anonymous.4open.science/r/Res-11B2.
>
> **Typos & Presentation:** Thanks. Following your suggestion, we will correct the typos and further improve the presentation.

---

> > ### Comment · Reviewer_6d3k · 2025-04-03
> >
> > I would like to thank the authors for these clarifications and providing the additional results. Could the authors clarify why the performance reported for the benchmark methods on the clean data seem to be significantly lower as the one reported in the original publications (i.e. Candy, Divide, and SCE-MVC)? Additionally, what is the intuition behind AIRMVC performing better than the baselines when no noise is present?
> >
> > While I certainly agree that it will be useful to follow the assumption of having one “ideal” view as a reference, simplifying the task. This assumption is more of a limitation in this case compared to the setup in prior work as you generally are aware if you have data or not, while the presence of noise in the data is more subtle. In addition, not having the view removes all the information in the view, while adding noise only degrades it. While I do not necessarily think that this is a major problem, I believe it would be a limitation worth discussing, potentially pointing to future work.

---

> > > ### Author Response · Authors · 2025-04-04
> > >
> > > **Explanation for experimental results:** Thanks for your comment. From the publicly available code of CANDY (line 11 of https://github.com/XLearning-SCU/2024-NeurIPS-CANDY/blob/main/dataset_loader.py) and DIVIDE (line 11 of https://github.com/XLearning-SCU/2024-AAAI-DIVIDE/blob/main/dataset_loader.py), it is evident that the datasets they used contain only two data views. In contrast, the datasets we employed, i.e., Caltech101 and Reuters, consist of five views. Therefore, the datasets used in our experiments are not the same. We directly report the performance obtained by reproducing their original code with our multi-view datasets, which accounts for the observed differences.
> > >
> > > Regarding SCE-MVC (https://openreview.net/pdf?id=xoc4QOvbDs), we used different clustering metrics, i.e., ACC, NMI, and PUR for AIRMVC, whereas SCE-MVC employs ACC, NMI, and ARI. Since the authors of SCE-MVC have not released their code, we reproduced their results based on the descriptions in their paper, which introduced some discrepancies. The experimental results demonstrate that AIRMVC achieves promising performance in the clean setting, rather than necessarily achieving SOTA performance, which aligns with our previous response.
> > >
> > > **Explanation for clean performance:** Thanks for your comment. From our reported results, AIRMVC demonstrates only promising performance in clean scenarios. Moreover, it does not achieve SOTA performance on some datasets. We further analyze the reasons behind its guaranteed performance. Compared with other modules, we design a contrastive learning mechanism to enhance the model's discriminative ability. Specifically, we employ a high-confidence threshold to improve the quality of positive and negative sample pairs in contrastive learning. Furthermore, we provide a concise theoretical analysis to justify the design of our contrastive learning mechanism.
> > >
> > > **Core idea of AIRMVC:** The core idea of AIRMVC is to explore the noisy problem in unsupervised multi-view scenarios. The experimental results in the submitted version demonstrate the effectiveness of AIRMVC in noisy scenarios. Although AIRMVC may not achieve SOTA performance across all datasets in the clean scenario, its promising performance could demonstrate its generalizability.
> > >
> > >
> > >
> > > **Future work:** Thanks for your comment. Noisy views are a prevalent challenge in real-world multi-view scenarios. However, existing research in MVC has largely overlooked this issue, and there remains a lack of standardized methodologies for simulating noisy datasets. In AIRMVC, we provide an **initial exploration** of the noisy view problem in an unsupervised setting. We are delight that our method of using an "ideal view" as a reference has received your recognition. Identifying a suitable reference view in an unsupervised scenario and designing more realistic noisy view simulation strategies are promising directions for future research. We fully agree that this is a worthwhile topic of discussion, and following your insightful suggestions, we will continue to explore this problem in greater depth.
> > >
> > >
> > > According to this year's ICML policy, we are not permitted to engage in multiple rounds of discussion. please trust that we have carefully considered and made every effort to address the concerns you raised. We kindly hope our response addresses your concerns. We greatly appreciate the time and effort you have dedicated to reviewing our work!

---

### Official Review · Reviewer_qT2Q · 2025-03-12

**Overall Recommendation:** 4

**Summary:**

To mitigate the impact of noisy data on multi-view clustering models, this paper proposes a method capable of automatically identifying and correcting noise. Specifically, the authors reformulate noise identification as an anomaly detection problem. Then, they design a hybrid correction strategy to enhance model robustness. Extensive experimental results demonstrate the effectiveness of the proposed approach.

**Claims And Evidence:**

In the submitted version of the paper, the motivation for handling noise is clearly defined and illustrated in Figure 1. Additionally, the authors conduct experiments to verify that the presence of noise adversely affects multi-view clustering performance. The submitted version effectively clarifies the research problem.

**Essential References Not Discussed:**

The comparison algorithms in the paper are primarily from 2024, incorporating the latest research methods.

**Experimental Designs Or Analyses:**

In this paper, the authors conducted extensive experiments, including comparative analyses under different noise ratios, comprehensive ablation studies, and sensitivity analysis experiments. Additionally, the methods compared in Tables 1 and 2 are all from 2024, ensuring a fair and up-to-date evaluation.

**Methods And Evaluation Criteria:**

In this paper, the authors conduct comprehensive experiments on six widely used benchmark datasets. The experimental results demonstrate that the proposed method effectively mitigates the impact of noise on clustering performance.

**Other Comments Or Suggestions:**

I. The paper contains a large number of formulas, and the vast majority of definitions and explanations are in accordance with the standards. However, in Equation 8 on page 4, the formula is too large and extends beyond the page. It needs to be adjusted.
II. It is recommended to add more experimental details for the visualization experiments in Section 5.4.

**Other Strengths And Weaknesses:**

S:
I. The paper investigates novel methods to mitigate the impact of noise on models, which is a practical area of research.
II. From the submitted version, it is evident that the authors provide theoretical analysis and conduct extensive experiments.
III. The proposed method is clearly described, making it easy to follow.

W:
I. The authors have conducted detailed experimental validation; however, there is a lack of validation regarding the time and space consumption of the proposed method. I recommend that the authors add experiments to address.
II. Figure 2 presents the overall framework of the paper. In the upper part of view2, does the dark blue color represent noise? I suggest adding definitions and descriptions of the different colored data in the legend.
III. The authors divided the experimental section into four parts, with the fourth part being the sensitivity analysis of the parameters. This part is placed in Appendix A.3.3, but the appendix is labeled as RQ3 instead of RQ4, which needs to be corrected.

**Questions For Authors:**

See above.

**Relation To Broader Scientific Literature:**

Compared to previous studies, this paper proposes a more effective approach to handling noisy data. The experimental results further validate this conclusion.

**Theoretical Claims:**

In Appendix A.2, the authors provide a mathematical proof, which theoretically supports the proposed method and enhances the credibility of the study.

---

> ### Author Rebuttal · Authors · 2025-03-31
>
> **Experiments of time and space cost:** Thanks. Following your suggestion, we conducted time and space complexity experiments on the six used datasets with 10% noisy ratio. Specifically, we measure the training time per epoch for all baselines using seconds as the evaluation metric. The space cost experiments are conducted on an NVIDIA A6000 GPU, measured in gigabytes (GB). The results are presented in Tab.1 and Tab.2. From these results, we observe that the time and space costs of AIRMVC remain within an acceptable range. In summary, AIRMVC demonstrates promising clustering performance while maintaining a reasonable computational cost.
>
> Tab.1 Time cost for AIRMVC.
>
> | Methods |              | BBCSports |  WebKB  | Reuters | UCI-dight | Caltech101 |  STL10  |  Avg.  |
> |:------------:|:------------:|:---------:|:-------:|:-------:|:---------:|:----------:|:-------:|:------:|
> |     CANDY    | NeurIPS 2024 |  0.0657   | 0.1206  | 0.2861  |  0.3325   |   3.2500   | 3.6200  | 1.2792 |
> |     RMCNC    |   TKDE 2024  |  0.1536   | 0.5148  | 0.4785  |  0.8962   |   3.9800   | 5.6800  | 1.9505 |
> |    TGM-MVC   |  ACM MM 2024 |  0.1206   | 0.2546  | 0.5931  |  0.6752   |   4.4200   | 6.2805  | 2.0573 |
> |    SCE-MVC   | NeurIPS 2024 |  0.1521   | 0.2675  | 0.6428  |  0.6028   |   4.0255   | 6.0865  | 1.9629 |
> |     MVCAN    |   CVPR 2024  |  0.1525   | 0.2756  | 0.4429  |  0.7636   |   4.3580   | 6.6210  | 2.1023 |
> |    DIVIDE    |   AAAI 2024  |  0.0795   | 0.1568  | 0.3524  |  0.3326   |   3.1350   | 3.6248  | 1.2802 |
> |    AIRMVC    |     Ours     |  0.0825   | 0.1486  | 0.3058  |  0.3390   |   3.0800   | 3.5200  | 1.2460 |
>
> Tab.2 Space cost for AIRMVC.
>
> | Methods |              | BBCSports | WebKB | Reuters | UCI-dight | Caltech101 | STL10 | Avg. |
> |:------------:|:------------:|:---------:|:-----:|:-------:|:---------:|:----------:|:-----:|:----:|
> |     CANDY    | NeurIPS 2024 |    1.91   |  1.79 |   2.11  |    2.21   |    2.36    |  3.03 | 2.24 |
> |     RMCNC    |   TKDE 2024  |    1.88   |  2.55 |   1.96  |    2.16   |    2.46    |  2.99 | 2.33 |
> |    TGM-MVC   |  ACM MM 2024 |    1.47   |  1.64 |   2.06  |    2.20   |    2.54    |  2.35 | 2.04 |
> |    SCE-MVC   | NeurIPS 2024 |    1.57   |  1.72 |   2.30  |    2.26   |    2.70    |  2.45 | 2.17 |
> |     MVCAN    |   CVPR 2024  |    1.56   |  1.57 |   1.66  |    1.28   |    1.44    |  1.47 | 1.50 |
> |    DIVIDE    |   AAAI 2024  |    2.02   |  1.78 |   2.05  |    2.19   |    2.36    |  2.97 | 2.23 |
> |    AIRMVC    |     Ours     |    1.60   |  1.63 |   1.71  |    1.34   |    1.55    |  1.48 | 1.55 |
>
> **Explanation for symbol in Fig.2:** Thanks. In Fig. 2(a), (b), and (c), the dark blue color represents noisy data. Following your suggestion, we will provide a more detailed explanation in the final version.
>
> **Typos & Format:** Thanks. Following your suggestion, we will revise RQ3 and Eq.8 in the final version and review similar issues to enhance the overall presentation.
>
> **Details for visualization experiments:** Thanks. We visualized the latent space features extracted by the encoder from the UCI-Digit dataset using the t-SNE algorithm. The visualization was performed every 20 epochs for the first 200 epochs. The experiments were conducted on an NVIDIA A6000 platform. We will provide additional descriptions in the future.

---

### Decision · Program_Chairs · 2025-05-01

**Decision:**

Accept (spotlight poster)

**Comment:**

This paper investigates robust multi-view clustering in noisy scenarios. The reviewers acknowledge some strengths:

1）The research addresses an important problem in the field.
2）A novel robust deep contrastive method for noisy multi-view clustering is proposed.
3）The paper provides comprehensive theoretical analysis and sufficient experimental validation.

Following rebuttal and discussion, all reviewers unanimously recommend acceptance.